# Widespread theta synchrony and high-frequency desynchronization underlies enhanced cognition

E.A. Solomon [1], J.E. Kragel[2], M.R. Sperling[3], A. Sharan[4], G. Worrell[5], M. Kucewicz[5], C.S. Inman[6], B. Lega[7], K.A. Davis[8], J.M. Stein[9], B.C. Jobst[10], K.A. Zaghloul[11], S.A. Sheth[12], D.S. Rizzuto[2] & M.J. Kahana[2]

The idea that synchronous neural activity underlies cognition has driven an extensive body of research in human and animal neuroscience. Yet, insufficient data on intracranial electrical connectivity has precluded a direct test of this hypothesis in a whole-brain setting. Through the lens of memory encoding and retrieval processes, we construct whole-brain connectivity maps of fast gamma (30–100 Hz) and slow theta (3–8 Hz) spectral neural activity, based on data from 294 neurosurgical patients fitted with indwelling electrodes. Here we report that gamma networks desynchronize and theta networks synchronize during encoding and retrieval. Furthermore, for nearly all brain regions we studied, gamma power rises as that region desynchronizes with gamma activity elsewhere in the brain, establishing gamma as a largely asynchronous phenomenon. The abundant phenomenon of theta synchrony is positively correlated with a brain region's gamma power, suggesting a predominant low-frequency mechanism for inter-regional communication.

[1] Department of Bioengineering, University of Pennsylvania, Philadelphia, PA 19104, USA. [2] Department of Psychology, University of Pennsylvania, Philadelphia, PA 19104, USA. [3] Department of Neurology, Thomas Jefferson University Hospital, Philadelphia, PA 19107, USA. [4] Department of Neurosurgery, Thomas Jefferson University Hospital, Philadelphia, PA 19107, USA. [5] Department of Neurology, Department of Physiology and Bioengineering, Mayo Clinic, Rochester, MN 55905, USA. [6] Department of Neurosurgery, Emory School of Medicine, Atlanta, GA 30322, USA. [7] Department of Neurosurgery, University of Texas Southwestern, Dallas, TX 75390, USA. [8] Department of Neurology, Hospital of the University of Pennsylvania, Philadelphia, PA 19104, USA. [9] Department of Radiology, Hospital of the University of Pennsylvania, Philadelphia, PA 19104, USA. [10] Department of Neurology, Dartmouth Medical Center, Lebanon, NH 03756, USA. [11] Surgical Neurology Branch, National Institutes of Health, Bethesda, MD 20814, USA. [12] Department of Neurosurgery, Columbia University Medical Center, New York, NY 10032, USA. Correspondence and requests for materials should be addressed to E.A.S. (email: esolo@pennmedicine.upenn.edu) or to M.J.K. (email: kahana@psych.upenn.edu)

The brain gives rise to behavior and thought through the coordinated activity and transfer of information between disparate regions[1]. Despite over a century of investigation into the brain's interconnectedness[2], the nature of these inter-regional interactions remains unknown. Our understanding of connectivity in the brain originates from studies that use indirect measures of neural activity, like blood–oxygen-level dependent (BOLD) functional MRI, extracranial electroencephalography (EEG), and magnetoencephalography[3]. While these techniques provide a useful picture of how distant brain regions act in concert during cognition, they lack the spatial or temporal precision of direct electrical recordings in the brain[4]. Until recently, the limited availability of such intracranial data made it difficult to assess the connectivity dynamics of the whole brain as it performs cognitive tasks.

Recent studies using direct brain recordings in neurosurgical patients have made it possible to robustly investigate neural synchronization, the coordinated activity of ensembles of neurons in different parts of the brain. Synchronization is an appealing mechanism for explaining how the brain stores memories, processes sensory inputs, or performs any operation that involves interlinking representations of the outside world[3], and it generally

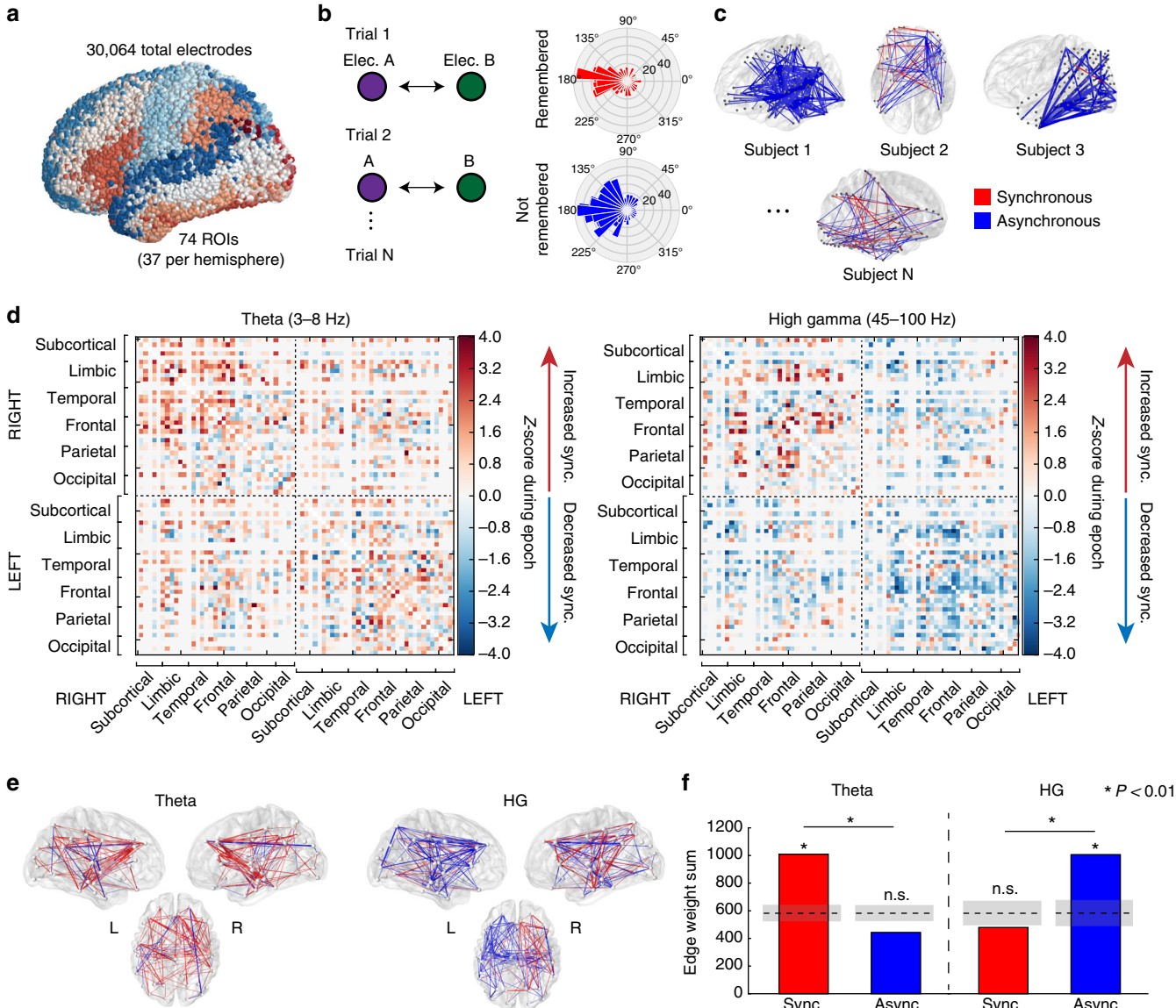

Fig. 1 Network construction and basic analysis. a 3D visualization of all surface electrodes included in our data set, colored by the Talairach atlas labels used in this article's analysis. b Schematic of spectral phase approach that compares the distributions of phase differences between electrodes across all trials of the verbal free-recall task. Significantly tighter distributions indicate greater synchronization. c Connectivity maps were extracted for each of 294 neurosurgical patients, reflecting the connectivity change associated with successful item recall. Effects were pooled across subjects and ROIs to construct the final network. Blue indicates decreased phase synchrony associated with successful encoding, red indicates increased synchrony. d 74 × 74 ROI adjacency matrices representing the z-scored time-averaged and frequency-averaged connection weights during the item presentation interval (0–1600 ms). The high gamma network is constructed from frequencies between 45–100 Hz, and theta from 3–8 Hz. Node indices are organized by lobe per the indicators on the axes. Gray areas represent connections between ROIs with fewer than seven subjects' worth of data. e 3D visualizations of the whole-brain HG and theta networks. f Summed positive and negative connection weights in each frequency band. In a remembered vs. not-remembered contrast, the total level of synchronous theta connections and asynchronous HG connections were significantly greater than chance (P < 0.01), and there was a significant frequency-synchrony interaction (P < 0.01, $\chi^2$ test). Dotted lines indicate mean chance level, shaded area ±1 STD

occurs on different timescales—or frequencies—of neural activity. In particular, gamma-band (30–100 Hz) synchronization is frequently invoked as a means for the brain to communicate between regions, since the fast nature of an oscillatory gamma signal is timed appropriately for rapid perceptual operations or induction of synaptic strengthening[5–8]. Support for this idea comes mostly from animal studies[6,7,9,10], though some human EEG studies also report cognitively induced low-gamma and short-range synchronicity[11–13]. However, others have argued that this body of work is conceptually and empirically deficient to defend the broad notion that high-frequency activity supports a meaningful neural connection[14–18]. Notably, conduction delays between cortical areas would make the precise synchronization of gamma oscillations difficult, and overall power at high frequencies may be too weak to support neuronal synchrony. Furthermore, the literature on this subject is mixed—even some of the most influential studies of gamma synchrony in humans report significant periods of desynchronization[11,12,19] and steep drop-offs in synchrony at higher frequencies[13]. These critiques raise the possibility that gamma does not serve to support communication between cortical regions, though this hypothesis has not been directly tested.

If activity in the gamma range is not synchronous, it may instead reflect the aggregation of rapid, stochastic firing in a population of neurons near an electrode, not an oscillatory modulation of activity that indicates coordinated activity across space[20,21]. Were this true, the general neural activation of a brain region—captured by the spectral power recorded at a cortical electrode—would rise as the synchronicity of that region with others would tend to fall. However, this form of broadband asynchronous activity may coexist with narrowband synchronous oscillations[22,23], and both may contribute to spectral changes at frequencies in the gamma band. In this case, it remains untested whether the oscillatory component of a gamma-band signal underlies long-range synchronization, and to what extent high-frequency activity during cognition reflects synchronous oscillations vs. asynchronous broadband activity.

If high-frequency activity is not the principal mediator of inter-regional synchronization, low-frequency interactions may be a promising alternative. Synchrony in the slower theta-band (3–8 Hz) has been reliably found to correlate with cognition in humans and animals[24–27], and theta oscillations are also linked to modulations of gamma activity[28,29]. However, low-frequency networks have not been characterized on a brain-wide scale, making it difficult to differentiate general principles of brain function from dynamics that may be particular to specific structures. It is possible that canonical regions such as the medial temporal lobe (MTL) and prefrontal cortex participate in low-frequency networks while less well-studied regions break from this trend. Moreover, low-frequency interactions have not yet been directly related to modulations of spectral power on a brain-wide scale, though probing these interactions may reveal the relationship between a region's functional connectivity and local processing.

In this study, our goal is to determine what principles underlie how neural activity is coordinated across the brain during memory processing, and to answer how spectral power and synchrony are related: To what extent is inter-regional communication mediated by low-frequency vs. high-frequency interactions? As the local high-frequency activity of a region increases, does its synchrony concomitantly decrease? How often do we observe high-frequency oscillations during cognition, and are they associated with long-range connectivity? While 294 subjects perform memory encoding and retrieval tasks—processes that rely on the integration and binding of information—we record intracranial electroencephalographic (iEEG) data and construct whole-brain networks of high-frequency and low-frequency phase interactions. To determine how synchrony changes over time and space, we parse these networks with graph-theoretic tools that identify hubs of the network, and then correlate the spatio-temporal pattern of synchrony at these hubs with simultaneously measured spectral power. Though our focus is on gamma-band and theta-band synchrony, we consider whether connectivity dynamics in these bands are better captured by broader frequency ranges, such as broadband low (< 30 Hz) and broadband high (> 30 Hz). We observe widespread desynchronization of high-frequency activity and synchronized low-frequency activity during memory processes, which correlate with regions of enhanced high-frequency power. Our findings support the notion that macroelectrode scale recordings largely reflect asynchronous neural firing at high frequencies, but also suggest a low-frequency mechanism for inter-regional communication.

## Results

**Quantification of brain-wide connectivity phenomena.** To assess connectivity between brain regions, we collected iEEG data from 294 patients undergoing clinical monitoring for seizures while they performed a verbal free-recall memory task (Fig. 1a; see Supplementary Fig. 1 for behavioral results). In this task, patients saw a series of words, each presented briefly on a screen, and were instructed to recall as many as possible. To construct networks of activity, we adopted a common spectral phase synchronization approach to measure connectivity between pairs of electrodes, called the phase-locking value, which quantifies the consistency of phase differences at a given frequency across trials of the experiment[30] (Fig. 1b, c; Methods section). In this paper, we primarily focus on regularly spaced frequencies in the 45–100 Hz range, referred to collectively as "high gamma," though we make no prior assumption as to whether these frequencies capture predominantly broadband asynchronous or oscillatory synchronous effects. Some analyses are extended to the 30–60 Hz range, referred to as "low gamma."

We first sought to quantify the grand-average modulation in high gamma (HG) and theta connectivity during item encoding that correlates with subsequent successful recall of that item—in other words, the relative level of synchronization comparing successful to unsuccessful encoding events. To measure this, we averaged the modulation in HG or theta connectivity across all possible electrode pairs that spanned every pair of anatomically defined regions of interest (ROIs) in all subjects (ROIs are based on automated Talairach atlas labeling[31], e.g., superior frontal gyrus, middle temporal gyrus, etc. See Methods section for details; Supplementary Table 1 for ROI abbreviations used in this paper). Connection weights are then z-scored against a null distribution, obtained by permuting remembered/not-remembered trial labels, to reflect the connection strength between ROIs relative to that expected by chance. The results of this procedure in the gamma and theta bands are adjacency matrices, which represent the pairwise connectivity between all ROIs (Fig. 1d), and which can be rendered as brain maps (Fig. 1e).

Encoding networks showed markedly different properties between HG-band and theta-band frequencies. As measured by the summed connection weights across the entire network, HG asynchrony and theta-band synchrony significantly correlated with successful encoding (Fig. 1f; $P < 0.01$ via permutation test of summed connection weights; Methods section). And though the network-wide level of synchronous activity in HG was not significant (permutation $P = 0.892$), this does not preclude the possibility that specific connections among ROIs are associated with successful memory encoding. Similarly, the overall level of

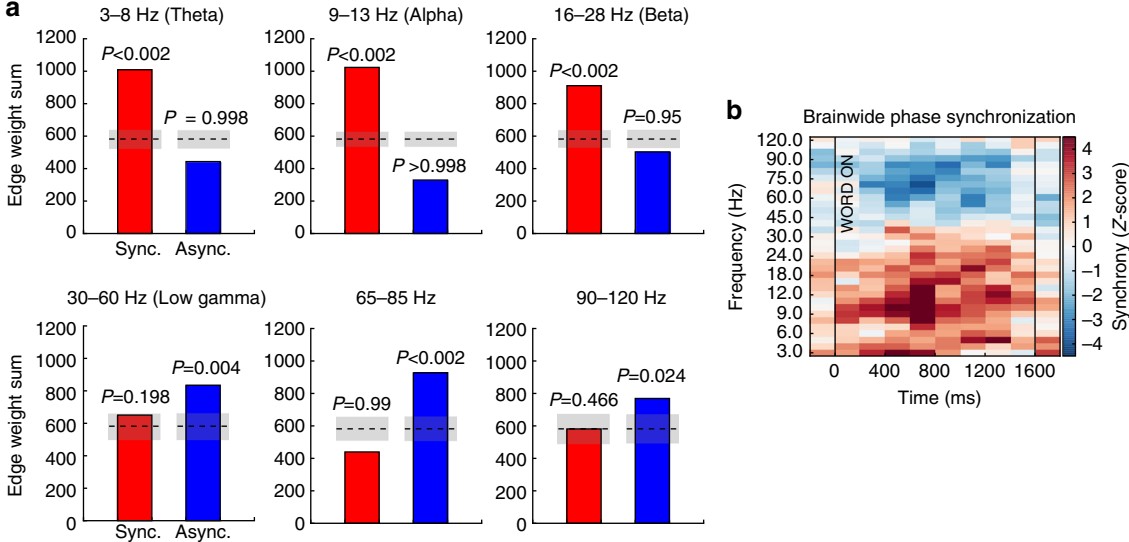

**Fig. 2** Synchrony effects from 3 to 120 Hz. **a** Overall level of SME synchrony/asynchrony in six frequency bands spanning 3 Hz to 120 Hz, measured as in Fig. 1f (edge weight sum). Shaded gray areas represent chance mean ±1 STD. **b** Z-scored brain-wide phase synchronization subsequent memory effect (SME) in the memory encoding interval, measured by summing all connection weights in the network, compared to the sum expected by chance. This analysis is performed in successive 200 ms windows spanning the encoding interval. Red reflects increased synchrony associated with successful memory encoding, blue reflects decreased synchrony (see Methods section for details). Vertical black lines indicate word onset and offset

theta-asynchronous interactions was not greater than chance (permutation $P > 0.99$). Extending this analysis to higher and lower frequencies revealed significant asynchrony (Fig. 2a; permutation $P < 0.05$) in frequencies between 30 Hz and 120 Hz, including the typical 30–60 Hz low gamma band. Significant synchrony in frequencies between 3 Hz and 28 Hz—theta, alpha, and beta bands—was also observed (permutation $P < 0.01$). The brain-wide connectivity z-score is given as a heatmap for each assessed frequency and timepoint in Fig. 2b.

Our findings of brain-wide HG asynchrony and theta synchrony during successful memory encoding suggest that low-frequency connections support information integration or coordinated brain activity during memory formation. However, we must first answer two deeper questions to determine whether there is a relationship between the neural activity of a region and the state of its connections to the rest of the brain. First, what is the brain-wide spatio-temporal pattern of synchrony/asynchrony during memory processing, and how does it relate to the pattern of local spectral power? Second, are there differing fundamental sources of neural activity that may have different power-synchrony relationships?

**Identification of network hubs.** Given that we observed significant levels of synchrony or asynchrony in low and high-frequency ranges, we next asked whether there is anatomic specificity to these phenomena. Are positive and negative connections homogenously distributed throughout the brain, or are there specific regions that exhibit greater modulation of connectivity during successful memory encoding?

To determine the most highly connected (or highly disconnected) ROIs, we turned to basic principles of graph theory. We used the node strength statistic (the sum of the unthresholded weights of every connection to a given node, here defined as an ROI) to identify which brain regions act as highly connected "hubs" in the memory network during the word presentation interval (0–1600 ms), the epoch with the greatest task-related modulation[32]. We defined hubs as ROIs with significantly greater node strength than expected by chance ($P < 0.05$ via permutation test of node strengths, Benjamini–Hochberg corrected for

multiple comparisons across ROIs), and we performed this analysis to identify hubs from all synchronous and asynchronous connections separately (Methods section). In HG, we found 3 synchronous hubs and 19 asynchronous hubs, which reflect brain regions that significantly increase or decrease their overall connectivity when a word is successfully encoded ($0.006 < P < 0.033$, FDR-corrected). The theta network exhibits 32 synchronous hubs widely dispersed across the cortex ($0.005 < P < 0.049$, FDR-corrected), but no hubs of asynchronous activity. Theta and HG hubs are depicted in Fig. 3, along with their strongest connections ($Z > 2.5$).

Taken together, these findings demonstrate that frontal, temporal, and MTL cortical regions became desynchronized from each other in HG during memory encoding. A smaller subset of right mesial frontal regions expressed synchronous activity with each other and functionally connect to temporal and parietal cortex. In the slower theta rhythm, the brain exhibited generally correlated activity, with numerous fronto-temporal, temporal-parietal, and interhemispheric functional connections.

Our finding that there is widespread theta synchronization during memory encoding follows from prior scalp and intracranial studies, which have shown that low-frequency entrainment is associated with cognition[24–27,33]. These findings also mirror findings in the fMRI literature of low-frequency networks that converge on the MTL in memory tasks[34]. The emergence of bilateral MTL as asynchronous hubs in HG is more surprising—this observation suggests, in a general sense, that structures such as the hippocampus do not synchronize at high frequencies with many other brain regions during successful encoding.

**Temporal modulation of connectivity effects.** To better characterize the role of these hubs in memory encoding, we asked whether a hub's participation in the HG or theta network changes over time. We assessed this by computing the node strength statistic at each 200 ms non-overlapping time window spanning 200 ms prior to 200 ms after the word presentation interval (Methods section). ROIs exhibited their strongest modulation of network participation between 400 ms and 1200 ms after onset of a word, with a particularly robust decrease in HG connectivity of

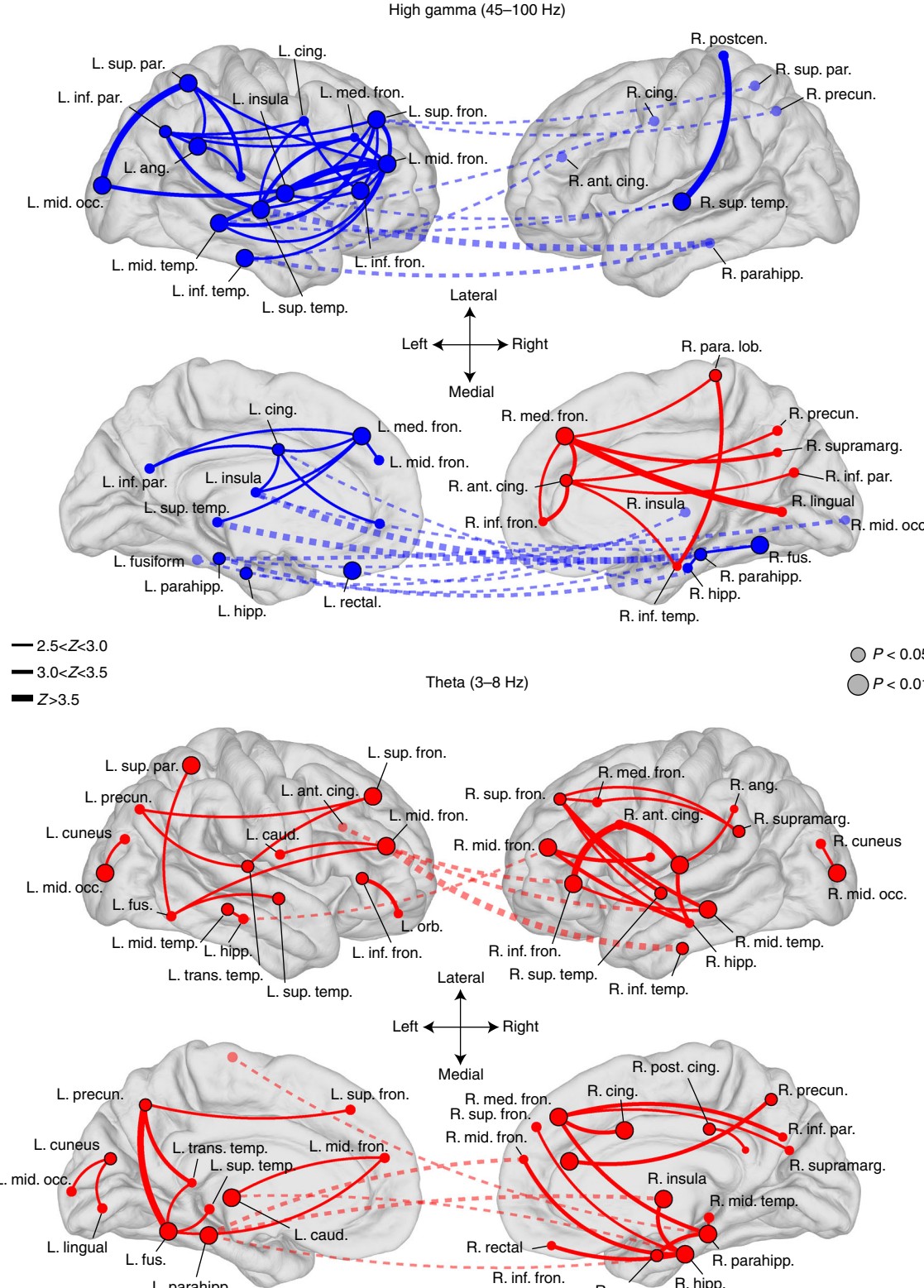

**Fig. 3** Network hubs. Depiction of hub ROIs identified in the brain-wide theta and high gamma memory encoding networks. The analysis was performed separately for all positive connection weights (red) and all negative connection weights (blue), yielding "synchronous hubs" and "asynchronous hubs," which respectively increase or decrease their connectivity with the network during successful memory encoding. Significant synchronous and asynchronous hubs for the item presentation interval are displayed according to their approximate localization on an average brain surface, with red circles indicating synchronous hubs and blue indicating asynchronous hubs (larger circles, FDR-corrected $P < 0.01$; smaller circles, $P < 0.05$). For each hub, the top five connections between that hub and any other part of the brain is plotted, if the connection weight $z$-score was greater than 2.5. Line thickness indicates absolute $z$-score value, according to the figure legend. Dashed lines indicate cross-hemispheric connections. Some labels are excluded from certain views to maintain readability. To aid visualization, hemispheres are reflected from their true position in the skull

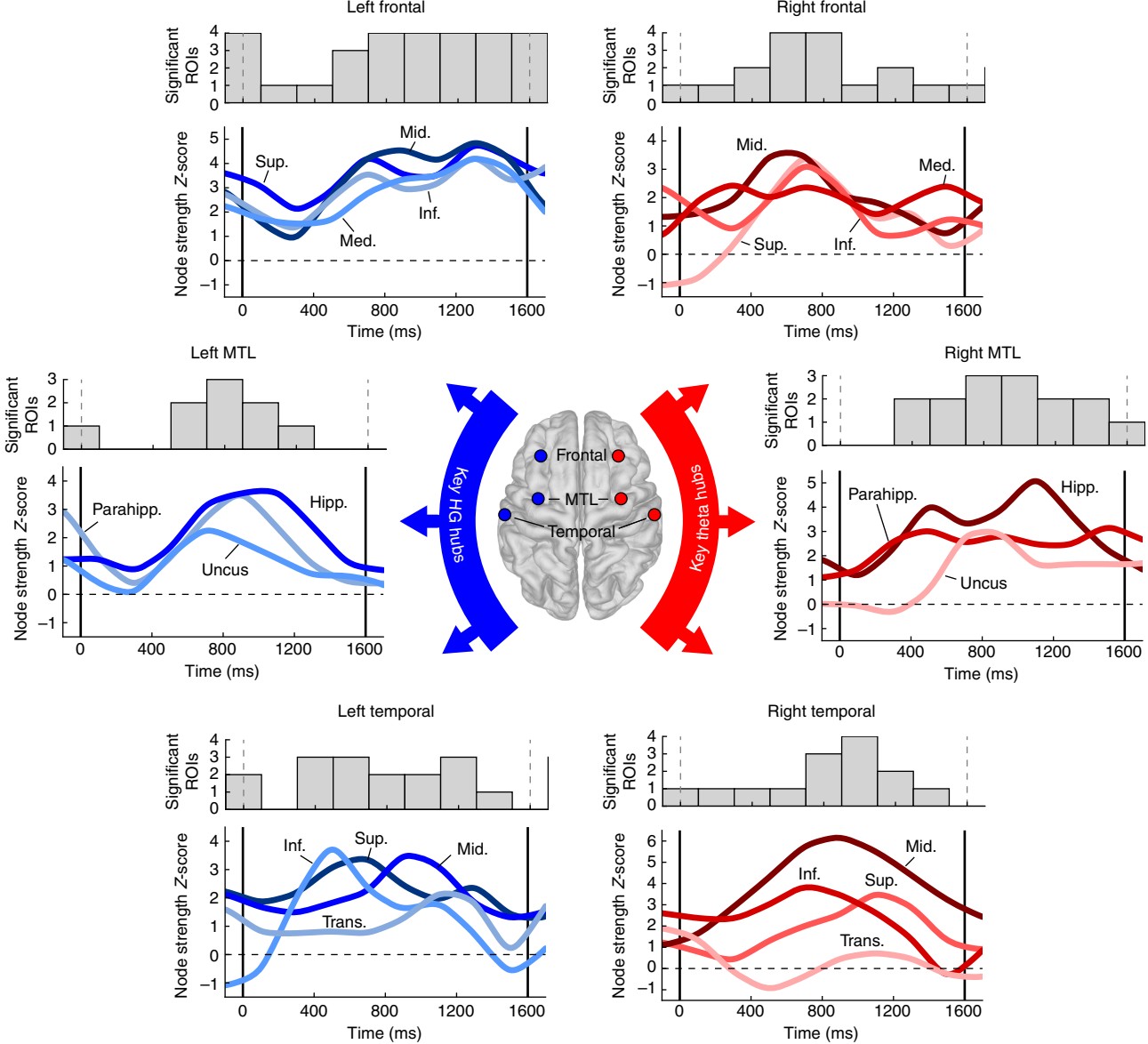

**Fig. 4** Timecourse of ROI participation in memory networks. Node strength as a function of time for 6 key regions that contain hubs in the theta or high gamma networks: right and left MTL, frontal lobe, and temporal lobe. Blue-shaded lines indicate asynchronous hub strength over time, while red indicates synchronous hub strength. Vertical lines indicate word onset and offset at 0 ms and 1600 ms. Above the z-scored timecourses are plotted the total count of specific ROIs within each broader region that reach significance at a given timepoint ($P < 0.05$). For visualization only, timecourses were smoothed with a 2-point moving average and radial basis filter

left MTL structures between 800–1000 ms (significant hippocampus, parahippocampus, and uncus ROIs, $P < 0.05$ via permutation test of node strengths; see Methods section for details). Correspondingly, the right MTL exhibited an increase in theta synchrony between 800–1200 ms (permutation $P < 0.05$). Theta synchrony in the right frontal lobe (significant middle, medial, inferior, and superior frontal cortices, permutation $P < 0.05$) peaked earlier, between 600–800 ms, while right temporal (significant middle, transverse, superior, and inferior temporal cortices, permutation $P < 0.05$) synchrony peaked between 1000–1200 ms (left cortical areas follow a similar pattern, see Supplementary Fig. 2). In Fig. 4, we show timecourses of node strength for ROIs in a subset of broader brain regions that contained hubs as identified previously (see Supplemental Fig. 2 for additional timecourses).

It is not surprising that we observed strong modulation of connectivity in both frequency bands during the item

presentation interval, since this time period is also known to feature the greatest change in spectral power[32]. It is unknown, however, how the directionality of connectivity changes relates to changes in spectral power—does enhanced theta synchrony or decreased HG synchrony in a brain region predict its HG or theta power?

**Relationship between connectivity and spectral power.** Having established the spatio-temporal dynamics of synchrony during performance of a memory task—noting the presence of MTL hubs that peak in their activity during the item presentation interval, for instance—we are now equipped to ask how these connectivity dynamics relate to spectral power, or the general neural activation of a region. Answering this question fills an important gap in knowledge about the nature of connectivity in

the brain, by showing how connectivity and power relate across a diverse array of cortical regions during memory processing.

We used the node strength of each ROI as a basis for a spectral power-synchrony correlation, asking whether a region's overall participation in the whole-brain network correlates with that

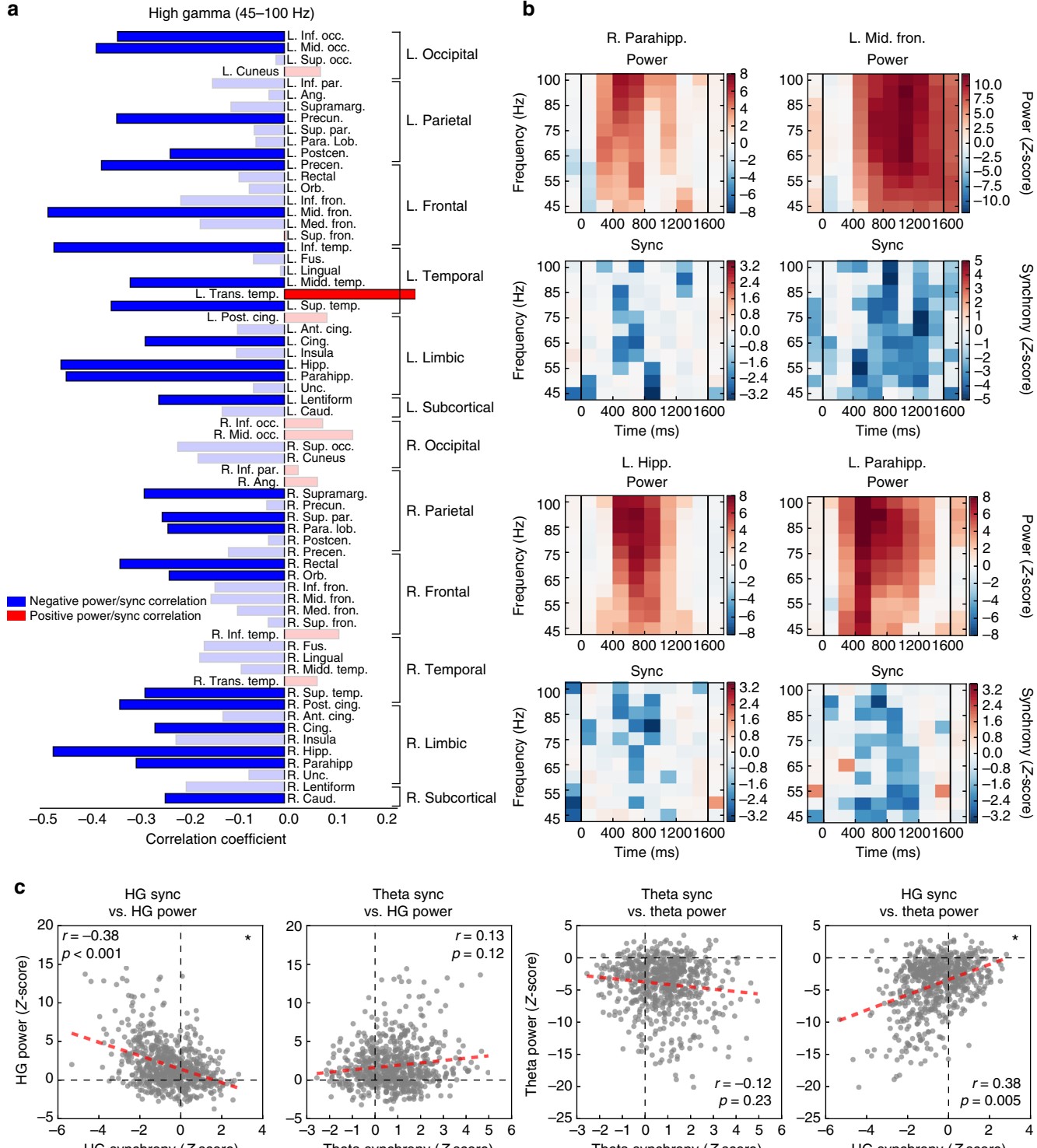

**Fig. 5** Power-synchrony correlations across the whole brain. **a** Pearson correlation of the modulation of high gamma node strength and power across time and frequency for each ROI, during the word encoding interval. Bar plots show the power-synchrony correlation for each ROI, with blue indicating negative and red indicating positive correlations. Faded bars are not significant after FDR correction for multiple comparisons ($\alpha = 0.05$). **b** For four example ROIs we depict time-frequency heatmaps of that ROI's z-scored spectral power (top) and z-scored node strength (bottom). Red colors indicate a relative increase of power/synchrony when an item is successfully encoded, while blue indicates a relative decrease. For visualization only, absolute z-scores < 1.5 are faded, and vertical bars indicate word onset and offset. **c** Pearson correlation of z-scored power and z-scored node strength (synchrony) against each other for all timepoints and all ROIs, after averaging within frequency band. HG power and HG synchrony are significantly inversely related ($P < 0.001$, permutation test), HG synchrony is positively correlated with theta power ($P = 0.005$), while other tested relationships do not meet significance

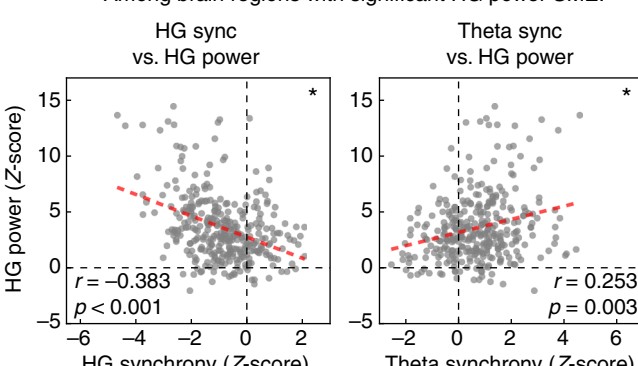

**a**  Among brain regions with significant HG power SME:

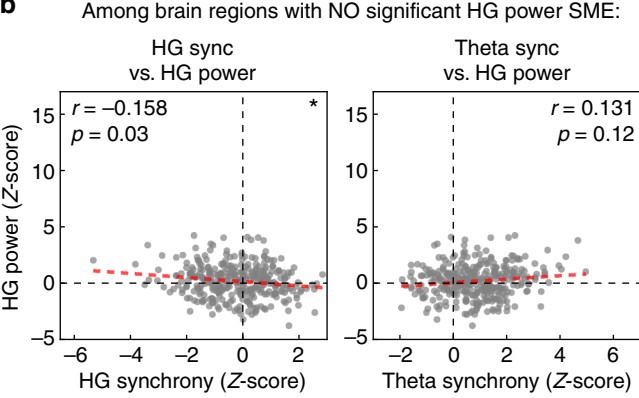

**b**  Among brain regions with NO significant HG power SME:

**Fig. 6** Correlations in the core memory network. **a** Power-synchrony correlations in the core memory network: the 37 ROIs with significant HG power subsequent memory effect (SME). **b** Power-synchrony correlations across the 37 ROIs with the no significant HG power effects. Among the core memory network—consisting mostly of left frontal, temporal, and MTL cortex—z-scored gamma power and z-scored synchrony were significantly anticorrelated, while theta synchrony and gamma power were significantly positively correlated (top row; $P < 0.001$ and $P < 0.01$ via permutation test, respectively). Among regions that did not exhibit strongly modulated HG activity in successful memory encoding, HG power and synchrony were still inversely correlated (Pearson correlation, $r = -0.158$, $P < 0.05$), but theta synchrony was not significantly predictive of HG power ($r = 0.131$, $P = 0.12$)

region's modulation of spectral power. For each ROI, we computed the power-synchrony (node strength) correlation across time and frequency in HG. We further asked how power and synchrony correlate across all ROIs and time after averaging effects within frequency band, enabling cross-band correlations.

We found that only one ROI exhibited a significant positive correlation between HG power and synchrony—the left transverse temporal gyrus—after Benjamini–Hochberg correction for multiple comparisons (Fig. 5a; Pearson correlation, $r = 0.27$, corrected $P = 0.017$). Twenty-four regions exhibited a significant negative correlation (Fig. 5a; Pearson correlation, $-0.48 < r < -0.23$, $4.6 \times 10^{-6} < P < 0.037$). Example power-synchrony heat-maps are given for four regions in Fig. 5b, depicting significant (corrected $P < 0.05$) negative correlations in the right parahippocampus, left MTL, and left frontal cortex.

Across all ROIs (74) and timepoints (10) together, the HG power-synchrony Pearson correlation was $-0.339$, $P = 0.002$ via a permutation test of synchrony and power correlation (Fig. 5c; see Methods section for details). In theta, within-ROI correlations showed 3 ROIs each of positive and negative power-synchrony relationships (corrected $P < 0.05$; Supplementary Fig. 3), but the general effect across all time and ROIs together was negative

though not significant (Pearson correlation, $r = -0.12$, permutation $P = 0.23$; Fig. 5c). Additionally, theta synchrony was weakly —but not significantly—correlated with HG power (Pearson correlation, $r = 0.11$, permutation $P = 0.2$; Fig. 5c). The brain-wide spectral power and synchrony at all frequencies from 3 Hz to 120 Hz are shown in Supplementary Fig. 4.

Measuring correlations across all ROIs together may obscure meaningful relationships within the subset of ROIs that actively participate in memory processing. We therefore sought to assess whether regions of the "core" memory network—those ROIs that significantly modulate their neural activity during successful memory encoding—exhibit power-synchrony dynamics that are different from the rest of the brain. These regions are said to exhibit a subsequent memory effect (SME)[35]. We found a total of 37 ROIs with no significant difference between HG power during successful vs. unsuccessful encoding, and classified these as outside the core memory network. Next, we matched these ROIs against the 37 ROIs with the largest SMEs, representing the core memory network (see Supplementary Table 1 for ROI classifications and z-scores). Among these two ROI subsets, we again computed power-synchrony correlations across all regions and all timepoints during the word encoding interval. In both groups, HG power and synchrony were inversely correlated (Fig. 6; Pearson correlation, $r = -0.38$ in-network and $r = -0.158$ out-of-network, $P < 0.001$ and $P < 0.05$ via permutation test; Methods section). However, only in the core memory network was theta synchrony significantly predictive of HG power (Pearson correlation, $r = 0.25$, permutation $P = 0.003$; Methods section). The difference in correlation between in-network and out-of-network does not reach significance (permutation $P = 0.20$).

**Generalization of network phenomena to memory retrieval**. To establish whether memory retrieval is also characterized by desynchronized HG activity and synchronized theta-band activity, we identified all of the 500 ms time windows in each subject's recall period that precede onset of a response vocalization, and compared connectivity dynamics against 500 ms time windows that are not followed by any vocalization for at least 2 s ("unsuccessful memory search"). Procedures are otherwise identical to those described in Fig. 1 and Methods section—phase-locking values in successful retrieval are compared to unsuccessful memory search, and these differences are pooled across subjects and ROIs. The result is a whole-brain connectivity map that reflects how phase synchrony is correlated with successful memory retrieval vs. unsuccessful memory search (Fig. 7a, c).

We found that the same network-level patterns of connectivity held true in the retrieval contrast compared to the encoding contrast. The HG retrieval network is characterized by a significant degree of asynchronous activity (Fig. 7b; $P < 0.01$ via permutation test of edge weight sum; Methods section) and an insignificant overall level of synchronous activity (permutation $P > 0.99$). In theta-band, there is a greater degree of synchronous activity compared to asynchronous (permutation $P < 0.01$; Fig. 7b). The relationship between power and synchrony also holds true in the analysis of recall. Even without sub-selecting for a core memory network as in Fig. 6, we find an inverse HG power-synchrony correlation (Pearson correlation, $r = -0.67$, permutation $P < 0.01$; Fig. 7d), although theta synchrony was positively but not significantly correlated with HG power (Pearson correlation, $r = 0.11$, permutation $P = 0.36$; Fig. 7d).

**Filtering for oscillatory activity**. Findings of HG desynchronization associated with successful memory encoding and retrieval suggest stochastic, non-oscillatory neural activity. However, it is possible that a mixture of two fundamental signals occupy the

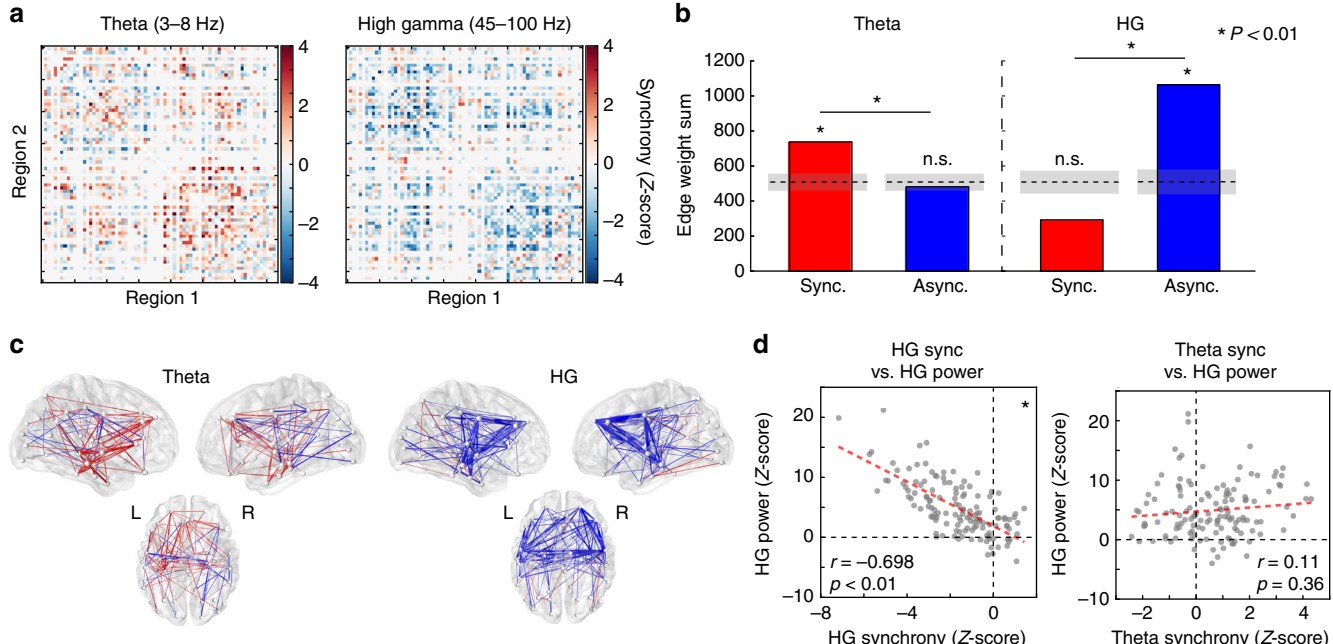

**Fig. 7** Generalization to memory retrieval processes. **a** Adjacency matrices, reflecting relative recall vs. baseline synchronization, organized as in Fig. 1d. **b** Summed positive and negative connection weights in each network, showing a strong desynchronization effect in gamma-band and a synchronization in theta ($P < 0.01$ for both). There was a significant frequency-synchrony interaction ($P < 0.01$, $\chi^2$ test). **c** 3D representation of gamma (top) and theta (bottom) retrieval networks, organized as in Fig. 1e. **d** Correlation of spectral power and phase synchronization across all regions (74) and timepoints spanning a retrieval trial (2). HG power and synchrony were significantly inversely correlated (Pearson correlation, $r = -0.698$, $P < 0.01$ via permutation test), while an ROI's theta synchrony was positively but not significantly predictive of HG power ($r = 0.11$, $P = 0.36$)

same frequency band: some components may be oscillatory, facilitating inter-regional communication, while others reflect asynchronous neural spiking activity. If the asynchronous component is much stronger or more commonplace than the oscillatory component, our results may be unable to capture true high-frequency synchronization that correlates with successful memory operations.

To answer whether high-frequency synchronization is driven by oscillatory dynamics, we examined which electrodes exhibit oscillations in the low gamma band ("LG," 30–60 Hz), utilizing a validated oscillation-detection routine ("Better Oscillation Detection" method, or BOSC, see Methods section for details; see Fig. 8a for an example)[36]. We identified the specific frequency and time at which a given electrode showed reliably increased oscillatory activity associated with trials that were later remembered, as compared to those forgotten ("oscillatory SME"; Fig. 8b for an example). Among the subset of electrodes with oscillatory SMEs, we reconstructed our phase synchronization networks to determine whether enhanced oscillatory activity was associated with increased inter-regional synchronization.

A total of 261 electrodes in our data set exhibited increased oscillatory activity associated with successful memory, maximally occurring at 52 Hz, between 400 and 600 ms after word onset (Fig. 8c). This is 1% of the total electrodes assessed; 5.6% of electrodes exhibited a 30–60 Hz SME and 10.4% of electrodes exhibited a 65–100 Hz power SME in the same time window (Fig. 8d). In the phase synchronization subnetwork that can be constructed from these 261 electrodes, we observed more synchronous ROI pairs than asynchronous pairs, when examining the network at the specific time (400–600 ms) and approximate frequencies (50–55 Hz) of maximal oscillatory SME (Fig. 8e; $P = 0.068$ via permutation test of edge count sum; Methods section). Examining this subnetwork at the same time but at higher frequencies (65–85 Hz) reveals a return to the

typical preponderance of asynchronous activity (Fig. 8e; permutation $P = 0.046$). This same trend can be observed in Fig. 8f, where maximal subnetwork-wide synchrony occurs in the same frequency range as that of maximal oscillatory SME (maximal $Z = 1.2$ at 55 Hz).

## Discussion

We set out to uncover fundamental principles that govern the electrophysiological networks of activity in the human brain. As 294 subjects performed a verbal free-recall memory task, we analyzed three frequency bands that have been strongly implicated in neural synchronization[37]: theta (3–8 Hz), low gamma (30–60 Hz), and high gamma (45–100 Hz). Gamma networks exhibited strong desynchronizations between brain regions, especially those that saw an increase in gamma power. Theta networks were characterized by enhanced synchrony, especially among regions with strong increases in HG power. Moreover, hubs of theta network activity tended to localize in frontal, temporal, and medial temporal cortices—regions that are known to play a strong role in memory encoding and retrieval[38].

Here we report findings that address whether theta or gamma band neural activity drives synchronization during memory processing. Gamma activity as a general biological mechanism of information transmission[3,5,6,24] is not backed by many compelling observations in the human brain. We found a profound decrease in HG synchronization that is associated with successful memory encoding and retrieval, especially among regions that see heightened overall HG activation. This relation is consistent with the hypothesis that broadband high-frequency activity in the human brain—as detected by macroelectrodes on the cortical surface—largely reflects the aggregation of fast, stochastic spiking activity of a population of neurons[15]. It refutes the notion that this kind of broadband signal synchronizes across long distances

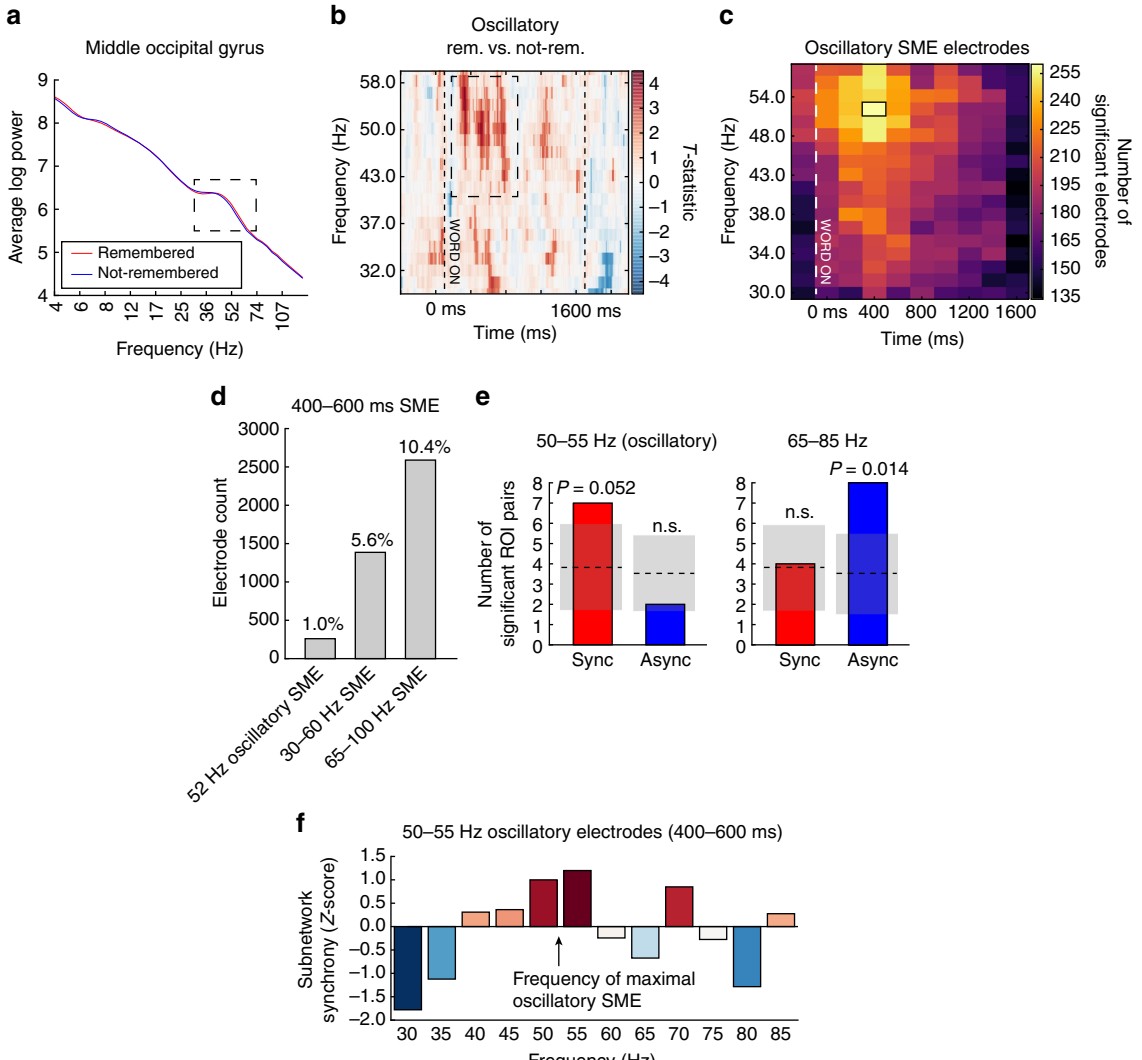

**Fig. 8** Synchronization of low gamma (30–60 Hz) oscillatory activity. **a** Example of an electrode exhibiting a gamma oscillation in the middle occipital gyrus, as detected by BOSC (see Methods section for details). Red line reflects average log power across all remembered events, blue line reflects average log power across not-remembered events. An isolated peak in the power spectrum is indicated, between approximately 36 and 74 Hz. **b** For the electrode in **a**, heatmap of the *t*-statistic reflecting the relative frequency of oscillations detected in remembered vs. not-remembered trials. Red colors indicate more oscillations detected at given frequency and timepoint in trials that were later remembered correctly. Vertical lines indicate word onset and offset (0 and 1600 ms). *T*-statistics < 2 are faded, for visualization only. Increased oscillatory power from 43 Hz to 58 Hz, coincident with the oscillatory peak in **a**, is indicated. **c** Count of all electrodes in the 294-subject data set that exhibit an oscillatory subsequent memory effect (SME) between 30 and 58 Hz, at each 200 ms epoch spanning the word presentation interval (Methods section). The most electrodes exhibit oscillatory SMEs at 52 Hz, between 400 and 600 ms after onset of a word (black box). **d** Count of electrodes in the data set that exhibit different kinds of SMEs between 400 and 600 ms: 52 Hz oscillation, 30–60 Hz average power, or 65–100 Hz average power. **e** Count of significant synchronous or asynchronous network connections using only the subset of electrodes exhibiting 52 Hz oscillatory SME at 400–600 ms. Synchronization effects were deemed significant at *P* < 0.05, with the chance mean and standard deviation at this significance level indicated in the gray shaded area. Left: counts observed at 50–55 Hz, near the frequency of maximal oscillatory SMEs (52 Hz). Right: counts observed in the 65–85 Hz range among the same electrode subset. The frequency/synchrony interaction is not significant (*P* = 0.11). **f** Average network synchrony (*z*-score) for the subnetwork of regions sampled in the 52 Hz oscillatory electrode subset, measured by summing the subnetwork connection weights at each frequency in the 400–600 ms window, and comparing to the sum expected by chance

during cognitive operations, though such interactions may still be at play in visual areas[39] and at smaller spatial scales.

Far from suggesting that brain regions are cut off from their neighbors, the observation that highly active memory regions significantly increase their theta synchronization offers a low-frequency mechanism by which the brain coordinates its many parts—a brain-wide finding that was suggested by prior studies which could only examine specific interactions[27,40]. Furthermore, our results demonstrate how theta networks exhibit time-varying structure, highlighting fronto-temporal hubs that strengthen their

connections starting 500 ms after onset of an item to be remembered. The fMRI connectivity literature parallels this, demonstrating broad, low-frequency networks that act to support human memory by convergence on the MTL[34,41–43]. The extent to which whole-brain iEEG-based networks overlap with fMRI networks is unexplored territory.

A small subset of electrodes exhibited increased narrowband gamma-oscillatory power associated with successful memory encoding. We observed increased long-range phase synchronization among this subset at the frequency of maximal oscillatory

activity. This indicates that, in some instances, gamma-band activity is organized into coherent, oscillatory waves that may serve to coordinate activity between different regions. The rareness of this phenomenon should not be understated; statistically reliable oscillatory SMEs were detected in only 1% of electrodes in our 294-subject data set.

A prior study by Burke et al.[40] in 2013 also suggested a general decrease in gamma synchronization and increase in theta during memory encoding in humans, but only at the level of lobe-wise interactions. The findings presented here extend that work in several important ways. First, we establish that decreases in synchrony accompany increases in high-frequency power, and that this fundamental relationship between power and synchrony manifests itself throughout the human brain. Second, we examined synchrony dynamics at a much finer spatial scale, allowing for the possibility that aggregation by lobe obscured synchronous gamma activity between nearby regions. Third, we teased out oscillatory effects in gamma, demonstrating that while synchrony is observed in rare instances, successful human cognition is overwhelmingly associated with a relative increase in asynchronous high-frequency signal.

Our data demonstrate the existence of two forms of gamma: Broadband asynchronicity, more common, and more rarely, narrow-band synchronous oscillations featuring long-range synchronization. Typically, the asynchronous broadband signal overwhelms the rare instances of oscillatory synchronization, explaining the widespread high-frequency desynchronization we found. These findings help reconcile a discrepancy in the synchronization literature. There is an established body of animal work in which cellular-scale recordings document high-frequency synchronization within or between inferotemporal, medial temporal, prefrontal, and occipital cortices during cognition[9,10,44–46]. But a far more tenuous corpus exists for humans at the macro-electrode scale—intracranial reports of synchronous gamma activity are rare and often simultaneously find significant periods of desynchronization[11,12,19]. Here, we quantified the extent to which human cognition is associated with two high-frequency neural dynamics, and found a predominant asynchronous signal at all frequencies above 30 Hz alongside a minority oscillatory synchronous signal. It is likely that more robust oscillatory activity can be detected with microelectrode recordings—as in prior animal work—but the results here speak against the importance of such dynamics at the scale of iEEG.

Whole-brain connectivity patterns still must be characterized in alternative memory paradigms, and other cognitive tasks altogether. Here, we investigated functional connectivity during a free-recall task, a prominent technique used to probe contextually mediated episodic memory. In freely recalling items from a previously studied list, subjects engage in a process of cue-dependent retrieval, wherein the cue for each recalled item includes information about the context of the target list and the previously recalled items. While this procedure disentangles neural activity from the influence of an external stimulus, experimenter-cued memory paradigms—especially cued recall and recognition—can provide additional valuable information about the timecourse of item retrieval.

The whole-brain connectivity network we report here extends the active frontier of network neuroscience[47]. By enabling the assessment of networks at different timescales of neural activity, whole-brain iEEG studies provide insights that go beyond non-invasive techniques—for example, the present study identified essentially opposite dynamics between low and high-frequency activity, which cannot be assessed by fMRI. By expanding our understanding of connectivity to the dimension of temporal frequency, this enhanced window into neural communication could reveal new ways in which network dynamics correlate with, or

even predict, disease states[48,49]. Connectivity maps also inform the use of direct brain stimulation as a therapeutic intervention—functional connectivity could serve as a model for predicting how stimulation effects propagate from one region to another, influencing activity throughout the brain[50]. If these connectivity-based models of brain function prove to be reliable, they may help clinicians use stimulation to repair the brain activity underlying damaged cognitive processes[51], such as memory deficits in patients with traumatic brain injury or neurodegenerative disease.

Distributed networks of electrical activity in the brain have remained largely uncharacterized despite their critical role in human cognition[6]. During memory encoding and retrieval, we discovered that whole-brain gamma networks were largely asynchronous, while theta networks were synchronous and specifically engaged among regions with a high degree of local processing. Our results lay the foundation for future study of low-frequency electrical networks as the primary driver of inter-regional communication in the human brain.

## Methods

**Participants.** A total of 294 patients with medication-resistant epilepsy underwent a surgical procedure to implant subdural platinum recording contacts on the cortical surface and within brain parenchyma. Contacts were placed so as to best localize epileptic regions. Data reported were collected at 10 hospitals over 14 years (2003–2017). Prior to data collection, our research protocol was approved by the Institutional Review Board at participating hospitals, and informed consent was obtained from the participants and their guardians.

**Free-recall task.** Each subject participated in a delayed free-recall task in which they studied a list of words with the intention to commit the items to memory. The task was performed at bedside on a laptop, using PyEPL software. Analog pulses were sent to available recording channels to enable alignment of experimental events with the recorded iEEG signal.

The recall task consisted of three distinct phases: encoding, delay, and retrieval. During encoding, lists of 12 words were visually presented in the native language (either English or Spanish) of the subject. Words were selected at random, without replacement, from a pool of nouns (http://memory.psych.upenn.edu/WordPools). Word presentation lasted for a duration of 1600 ms, followed by a blank inter-sitmulus interval of 750 to 1000 ms. Presentation of word lists was followed by a 20 s post-encoding delay. Subjects performed an arithmetic task during the delay in order to disrupt memory for end-of-list items. Math problems of the form A + B + C = ?? were presented to the participant, with values of A, B, and C set to random single digit integers. After the delay, a row of asterisks, accompanied by a 60 Hz auditory tone, was presented for a duration of 300 ms to signal the start of the recall period. Subjects were instructed to recall as many words as possible from the most recent list, in any order during the 30 s recall period. Vocal responses were digitally recorded and parsed offline using Penn TotalRecall (http://memory.psych.upenn.edu/TotalRecall). Subjects performed up to 25 recall lists in a single session.

A subset of 92 patients performed a variant of the previously described task. List presentation consisted of a total of 15 items. In addition, a green fixation cross served as a list-cue to signal an upcoming list of words. The list-cue was presented for a duration of 1600 ms, followed by the presentation of a blank screen for 800–1200 ms. The ISI in this variant of the task lasted from 800 to 1200 ms in duration. The recall period for this version of the task was 45 s in length.

**Electrocorticographic recordings.** iEEG signal was recorded using subdural grids and strips (contacts placed 10 mm apart) or depth electrodes (contacts spaced 5–10 mm apart) using recording systems at each clinical site. iEEG systems included DeltaMed XlTek (Natus), Grass Telefactor, and Nihon-Kohden EEG systems. Signals were sampled at 500, 512, 1000, 1024, or 2000 Hz, depending on hardware restrictions and considerations of clinical application. Signals recorded at individual electrodes were converted to a bipolar montage by computing the difference in signal between adjacent electrode pairs on each strip, grid, and depth electrode. Bipolar signal was notch filtered at 60 Hz with a fourth order 2 Hz stop-band butterworth notch filter in order to remove the effects of line noise on the iEEG signal.

**Anatomical localization.** Anatomical localization of electrode placement was accomplished using independent processing pipelines for depth and surface electrode localization. For patients with MTL depth electrodes, hippocampal subfields and MTL cortices were automatically labeled in a pre-implant, T2-weighted MRI using the automatic segmentation of hippocampal subfields multi-atlas segmentation method[42]. Post-implant CT images were coregistered with presurgical T1 and T2-weighted structural scans with Advanced Normalization Tools[52]. MTL depth electrodes that were visible on CT scans were localized within MTL

subregions by neuroradiologists with expertize in MTL anatomy[53]. Subdural electrodes were localized by reconstructing whole-brain cortical surfaces from pre-implant T1-weighted MRIs using Freesurfer[54]. ROIs used for connectivity analyses were given by the Talairach label of a given electrode's position after mapping final contact locations to Talairach space, with the exception of any electrode localized to a hippocampal subfield, which were collectively labeled "hippocampus." We considered 37 possible labels for each hemisphere, or 74 total.

In a subset of 92 patients, contact localization was accomplished by coregistering the post-operative CTs with post-operative or pre-operative MRIs using FSL (FMRIB Software Library) BET (Brain Extraction Tool) and FLIRT (FMRIB Linear Image Registration Tool) software packages.

Contacts placed in an epileptogenic area or in non-neural tissue (as determined by a clinician) were excluded from all the analyses in this report.

**Data analyses and spectral decomposition.** iEEG signals were all treated as bipolar montages (a difference in the raw signals from two adjacent electrodes), with sampling rates varying between 500 Hz and 2000 Hz, depending on the subject. We convolved the signal (downsampled to 500 Hz) from each bipolar electrode in each subject with complex-valued Morlet wavelets (wave number 5) to obtain phase and power information. We used 35 wavelets from 3–120 Hz, though most analyses focus on the 45–100 Hz (high gamma) and 3–8 Hz (theta) ranges (HG: 11 wavelets spaced 5 Hz, except between 90 Hz and 100 Hz; theta, 6 wavelets space 1 Hz). Each wavelet was convolved with 3600 ms of data surrounding each word presentation (referred to as "trial," 1000 ms before word onset to 2600 ms after word onset), and buffered with 1000 ms on either end (clipped after convolution).

For each subject, for all possible pairwise combinations of electrodes, we compared the distributions of phase differences in all remembered trials against all not-remembered trials, asking whether there is a significantly higher concentration, or tightness of the distribution, in one or the other (Fig. 1b). To do this, we found the difference of the mean resultant vector lengths (often called phase-locking value) of the remembered and not-remembered phase difference distributions ($\overline{R}$ values computed with Circular Statistics Toolbox[55]):

$$D_{pq}(f,t) = \overline{R}_{rem} - \overline{R}_{nrem}$$

Where $\overline{R}_{rem}$ and $\overline{R}_{nrem}$ refer to the mean resultant vector lengths of all remembered and not-remembered trials, $pq$ is an electrode pair, $f$ is a frequency band, and $t$ is a window in time.

Intuitively, a higher resultant vector length (which falls between 0 and 1) reflects a tighter distribution of phase differences and greater synchronization between two electrodes. Therefore, higher positive differences ($D$) indicate greater phase-locking for remembered trials, whereas lower negative differences reflect greater phase-locking for not-remembered trials. $D$ was computed for each frequency spanning a range from 3 to 120 Hz, and for 18 non-overlapping 200 ms time windows spanning the trial, by averaging phase difference values within those windows before computing phase-locking values and their corresponding $D$. Unless stated otherwise, the analyses in this report consider only the eight 200 ms windows between word onset (0 ms) and offset (1600 ms), called the "item presentation interval."

$\overline{R}$ values are biased by the number of vectors in a sample. Since our subjects generally forget more words than they remember (Supplementary Fig. 1), we adopt a nonparametric permutation test of significance. For each subject, and each electrode pair, the phase synchrony computation described above was repeated 500 times with the trial labels shuffled, generating a distribution of $D$ statistics that could be expected by chance for every electrode pair, at each frequency and time window. Since only the trial labels are shuffled, the relative size of the surrogate remembered and not-remembered samples also reflect the same $\overline{R}$ sample size bias. Consequently, the true $D$ ($D_{true}$) can be compared to the distribution of null $D$s to derive a $P$-value or $z$-score. Higher $z$-scores indicate greater synchronization between a pair of electrodes for items that are later recalled.

To construct a network of phase synchrony effects between all brain regions, we pooled synchrony effects across electrode pairs that span a pair of ROIs, and then pooled these ROI-level synchronizations across subjects with that pair of ROIs sampled (ROIs were determined by the Talairach label for each electrode after coregistration). To do this, we first averaged the $D_{true}$ values across all electrode pairs that spanned a given pair of ROIs within a subject. Next, we averaged the corresponding null distributions of these electrode pairs, resulting in a single $D_{true}$ and a single null distribution for each ROI pair in a subject. We then averaged the $D_{true}$ values and null distributions across all subjects with electrodes in a given ROI pair. By comparing the averaged $D_{true}$ to the averaged null distribution, we computed a $z$-score at each frequency and temporal epoch that indicates indicating significant phase synchrony or asynchrony, depending on which tail of the null distribution the true statistic falls. Higher $z$-scores indicate greater synchronization between a pair of ROIs for items that are later recalled.

**Network construction and analyses.** Using the population-level statistics described above, a 74-by-74 adjacency matrix was constructed for each of the 18 non-overlapping temporal epochs and for each frequency. This matrix represented every possible interaction between all ROI pairs. The $z$-score of the true $D$ relative

to the null distribution was used as the connection weight of each edge in the adjacency matrix. Negative weights indicate ROI pairs that, on average, desynchronized when a word was recalled successfully, and positive weights indicate ROI pairs that synchronized when a word was recalled successfully. We zeroed-out any ROI pairs in the adjacency matrix represented by less than 7 subjects' worth of data, to limit the likelihood that our population-level matrix is driven by strong effects in a single or very small number of individuals. 1243 ROI pairs (out of a possible total of 2701) were excluded due to low subject counts, comprised largely of interhemispheric pairs (795 pairs, or 64% of those excluded) and pairs involving regions where electrodes are less commonly placed, including basal ganglia and occipital cortex.

Since it is possible that collections of weaker connection weights may still account for significant structure in our network, we did not apply a $z$-score threshold before further analyses. To assess for the significance of phenomena at the network level, we instead used 500 null networks that can be constructed on the basis of $D$s derived from the shuffled trial labels to generate a distribution of chance network-level statistics. True statistics were compared to these null distributions to obtain a $P$-value or $z$-score (e.g., network-wide summed connection weights were computed for true and null networks and reported in Figs. 1f, 2a, and 6b).

Accordingly, for every operation performed on the true connectivity network, the same was done on each of the 500 null networks that reflect connection strengths expected by chance. For example, to ask whether a ROI has a significantly increased node strength at a given point in time (see subsection on Hub analysis), node strength was computed for each of the 500 null networks to generate a distribution of strengths expected by chance. The true node strength is compared to the null distribution in order to get a $Z$-score or $P$-value.

Adjacency matrices reflect the average connectivity strength during the item presentation interval (0–1600 ms) for each frequency band. To create them, we averaged true connection strengths within frequency bands, then averaged across the eight 200 ms time windows in this interval, and compared the result to the time/frequency average from each of the 500 null networks, resulting in a new $Z$-score for the time/frequency-averaged network (Fig. 1d).

**Hub analysis.** To identify which ROIs are more highly synchronous or asynchronous, we used the node strength statistic from graph theory to identify "hubs" of the network. Node strength reflects the sum of all connection weights to a particular node (or ROI) in the network, and is formalized as:

$$k_i^w = \sum_{j \in N} w_{ij}$$

Where $k$ is the node strength of node $i$, and $w_{ij}$ refers to the edge weight between nodes $i$ and $j$. $N$ is the set of all nodes in the network[47].

To identify hubs during the word presentation interval, we first averaged connection weights within a frequency band and across the presentation interval (as done in Fig. 1). Each ROI's node strength is then computed with these time/frequency-averaged weights, per the equation above. The same procedure was done for each of the 500 null networks generated from shuffled trial labels (see "Network construction and analyses"), creating a null distribution of node strength for each ROI. $P$-values were obtained by observing where a true node strength falls in its corresponding null distribution. Final $P$-values were corrected for multiple comparisons (Benjamini–Hochberg procedure, $\alpha = 0.05$ or 0.01) to yield the final tally of significant hubs. This process was done for all synchronous ($Z > 0$) and asynchronous ($Z < 0$) connections separately, yielding synchronous and asynchronous collections of hub ROIs. For visualization only, connections depicted in Fig. 3 were derived by ranking the time/frequency-averaged connection weights of each hub, and selecting up to the top 5 connections above a $Z$-score of 2.5.

To construct ROI activation timecourses, we compared the frequency-averaged node strength at each time window against its corresponding null distribution to generate a $Z$-score and a $P$-value, done separately for all positive and negative connection weights. Our selection of right and left MTL, frontal, and temporal cortices was driven by their implication in memory in prior literature[1,40,56] and the presence of gamma and/or theta-band hubs in each of those broad regions (Fig. 4).

**Power-synchrony analysis.** Spectral power was obtained by the same Morlet wavelet convolution as used to extract phase information (see "Data analyses and spectral decomposition"). For all bipolar electrodes in each subject, we log transformed and $z$-scored power within each session of the free-recall task, which comprises approximately 300 trials. Power values were then averaged into 8 non-overlapping 200 ms windows spanning the entire trial, matching our procedure for phase synchrony.

To assess the statistical relationship between power and later recollection of a trial word (called the SME), power values for each electrode, trial, time, and frequency were separated into two distributions according to whether the trial word was later remembered or not-remembered, and Welch's $t$-test was performed to compare the means of the two distributions. Next, we shuffled the trial labels 500 times and recomputed the $t$-statistic, reflecting power effects that could be observed by chance. The true $t$-statistics were averaged across all electrodes that occur in a given ROI, as are the null distributions, and those statistics are next averaged across all subjects with electrodes in that ROI. Finally, the averaged true $t$-statistic was

compared to the averaged null distributions to get a $z$-score at each time-frequency point for a given ROI. These $z$-scores are reported as a heatmap in Fig. 5b, and we find the Pearson correlation against node strength $Z$-scores as described in "Hub analysis" (i.e., a correlation across time-frequency pixels). Correlation $P$-values are then FDR-corrected for multiple comparisons (corrected $P < 0.05$).

To assess correlations across all time and all ROIs (Figs. 5c, 7d) or ROI subsets (Fig. 6), we first averaged $Z$-scored node strength and $Z$-scored power within each frequency band. Then, we correlated the strength and power values across all item presentation timepoints and all ROIs (i.e., each vector contains time windows by # ROIs total elements). To assess significance of these correlations, we adopted a permutation procedure that maintains the spatial and temporal dependency between data points: We assessed the power-synchrony correlation for each possible 1-shift of one vector against the other, and again for the mirror image of that vector. This procedure resulted in a distribution of chance correlations, against which we compared the true correlation to obtain a $P$-value.

**Retrieval analysis**. To find out whether principles of brain function uncovered in the memory encoding contrast generalize to different cognitive operations, we further analyzed connectivity in a retrieval contrast. This was done in a manner similar to Burke et al. 2014[57], as follows. For each subject, we identified any 500 ms interval during the recall period after which no response vocalization occurred for at least 2 s, and compared the neural activity in these "unsuccessful memory search" intervals to the 500 ms of activity immediately prior to successful item recollection. Phase difference values were averaged across two 250 ms time windows spanning these trials, as opposed to 200 ms windows in the encoding analysis. All other data analysis and spectral methods were matched exactly. This analysis was performed on a subset of 197 subjects with detailed retrieval-period information.

**Oscillations analysis**. We adopted a widely used method for oscillation detection, called P$_{episode}$ or BOSC[36,58,59] ("Better OSCillation detection"). Briefly, this method applies two criteria for the identification of a true oscillation: a minimum time (at least three cycles), and a significant deviation of spectral power from a robust linear fit to the log-frequency vs. log-power curve (a spectral "peak", see Fig. 8a for an example). For each timepoint and frequency assessed, BOSC indicates whether an oscillation is present under these criteria. For further details on BOSC implementation, see Hughes et al[36].

For each electrode in our data set, we used BOSC to find out whether the presence of low gamma (30–58 Hz, to minimize line noise artifact) oscillations was correlated with whether a word would later be remembered or forgotten ("oscillatory SME"). We used this range because at higher frequencies, the BOSC measure becomes less reliable as the log-frequency vs. log-power becomes nonlinear. For every trial, we computed the fraction of time occupied by an oscillation in each of eight 200 ms window spanning the 1600 ms item presentation interval, doing so for 18 log-spaced frequencies between 30 and 58 Hz. The result was a measure of oscillatory activity in each time/frequency pixel for each trial. We then grouped the trials by whether the word presented was later remembered or forgotten, and computed Welch's $t$ test between the remembered and forgotten distributions. $P$-values were FDR-corrected for multiple comparisons across time/frequency pixels ($\alpha = 0.1$, a deliberately liberal threshold to allow for enough electrodes to analyze pairwise synchronization). The count of electrodes with significant memory-correlated oscillatory power is depicted in Fig. 8c.

To determine whether an electrode exhibited an SME without directly assessing for oscillations (and thus capturing elevations in spectral power due to non-oscillatory activity), we computed an electrode's spectral power SME (described in "Power-synchrony analysis" above) averaged across 5-Hz spaced frequencies within the 30–60 Hz and 65–100 Hz bands at each of the eight 200 ms windows. $P$-values were FDR-corrected and declared significant at $\alpha = 0.1$, as above. The count of electrodes with significant SMEs at the 400–600 ms window is depicted in Fig. 8d.

Since the number of electrodes exhibiting gamma-oscillatory SMEs is small (approx. 260), the networks that can be constructed from that data set are sparse. Accordingly, the same procedure as described in "Data analyses and spectral decomposition" and "Network construction and analysis" is used on this small subset of electrodes (found in 44 subjects) to generate a map of some pairwise ROI synchronizations—a subnetwork—but not a whole-brain network. No threshold was applied on the number of subjects needed to contribute to an ROI pair. In Fig. 8e, subnetwork connection $z$-scores during the 400–600 ms window were tested for significance at the $P < 0.05$ level (uncorrected), and compared against the number of significant connections expected at that level by chance (i.e., shuffled trial labels). In total, 50–55 Hz were chosen as the closest frequencies to the frequency of maximal oscillatory SME (~52 Hz). In Fig. 8f, the $z$-scored mean subnetwork connection weight at 400–600 ms was plotted as a function of frequency.

**Data availability**. Raw electrophysiological data used in this study is freely available at http://memory.psych.upenn.edu/Electrophysiological_Data.

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

## Acknowledgements

We thank Blackrock Microsystems for providing neural recording equipment. This work was supported by the DARPA Restoring Active Memory (RAM) program (Cooperative Agreement N66001-14-2-4032), as well as National Institutes of Health grant MH55687 and T32NS091006. We are indebted to all patients who have selflessly volunteered their time to participate in our study. The views, opinions, and/or findings contained in this material are those of the authors and should not be interpreted as representing the official views or policies of the Department of Defense or the U.S. Government. We also thank Dr Danielle S. Bassett, Dr Youssef Ezzyat, and Dr Christoph Weidemann for providing valuable feedback on this work.

## Author contributions

E.A.S., J.E.K., M.J.K., and D.S.R. designed the study. E.A.S. and J.A.K. analyzed data, and E.A.S. wrote the paper. M.R.S., A.S., G.W., M.K., C.S.I., B.L., K.A.D., J.M.S., B.C.J., K.A.Z., S.A.S. recruited subjects, collected data, and performed clinical duties associated with data collection including neurosurgical procedures or patient monitoring.

## Additional information

**Competing interests:** The authors declare no competing financial interests.

