## [Peer Review File · Nature Communications]

Reviewers' comments:

Reviewer #1 (Remarks to the Author):

The manuscript by Solomon et al. seeks to understand patterns of brain-wide interactions during memory processing using an extremely large data set of direct local field potential recordings from the human brain. The manuscript seeks to test an important hypothesis: are behavioral related changes in memory encoding and retrieval related to higher frequency (gamma band) or lower frequency (theta band) interactions in humans? This issue has been contentious, as the authors rightly argue, because past studies have made strong claims about the importance of gamma synchrony to human cognition (and memory) more generally. At the same time, a growing and rich literature has shown that lower frequency (theta) oscillations instead serve as the foundation for interregional communication. This paper helps resolve this important debate for the first time by directly comparing synchrony, asynchrony, and local oscillatory patterns across multiple sites in the human brain. Using a multivariate, graph theory approach. The authors find that higher frequency oscillations tend to desynchronize and lower frequency oscillations tend to synchronize. The authors identify several important hubs, including the hippocampus, that underlie these effects during encoding and retrieval, and describe their temporal properties, which vary by region depending on when the word is being studied or retrieved. The authors find little correlation between gamma desynchronization and local power across the whole brain, and similarly, there was little correlation between low frequency theta and theta synchrony. However, when focusing on ROIs showing subsequent memory effects (parts of the "recollection" network), increases in theta synchrony did correlate with increases in gamma power, providing an important link between local processing and globally synchronous activity.

Overall assessment: This is an extremely important and well-considered study that will have a significant impact on the field. This is true for many reasons. For one, the study is the first to directly test the "gamma vs. theta" synchrony hypothesis, decidedly resolving this debate clearly in the favor of theta. This helps resolve decades of debate on this topic. Another important component of the study is the identification not only of hubs that in turn connect to the memory literature (i.e., MTL), but their specific time courses. Additionally, the study nicely ties together disparate work that has typically treated encoding and retrieval separately and links them under the same rubric of network coherence. Finally, the study is high-powered and the experimental paradigm both robust yet ecologically valid and helpful. I have only minor concerns aimed at improving the breadth and impact of the findings, which I detail below.

1. Methods: The authors indicate that they use bipolar montages. This makes sense for analysis of local power but it is less clear how this is optimal for looking at global synchrony. In fact, many past papers testing such issues have used an averaged reference. Do the results generally hold if the authors use an averaged reference, based on areas that do not include ictal or inter-ictal activity?

2. Methods: The use of "F statistic" is unclear. What is this referring to? Why were they

summed?

3. The N varies widely for different electrode placements. The extensive coverage is a major strength of the paper. Yet, the inconsistent placement across subjects, ranging from 7-200 subjects overlapping with the same area, is a potential weakness. Can the authors perform an analysis in which they (roughly) equate subject number, perhaps randomly choosing subjects so N=20, to show that the results still hold?

4. In many places (Figure 1F, Figure 6F), the authors imply an interaction effect but do not directly test this. Can they use a Chi Square test of association to explicitly test for a cross-over interaction effect based on edge weight?

5. Abstract: the authors state "Yet, the absence of high-resolution data on whole brain electrical connectivity during cognitive tasks has prevented a direct test of this hypothesis." This statement needs amending. In fact, numerous other intracranial studies have used multilobular grids and depth electrodes to look at memory, and cognition more generally. I think what the authors are trying to argue is that their coverage, because of their large sample size, is more extensive. This sentence should be rephrased.

6. p. 11: "This relation is consistent with the hypothesis that broadband high-frequency..."

I think the authors intended to state that gamma is a reflection, in part, of spiking activity as well as synaptic activity. If they add "in part" to the phrasing of the sentence "reflects, in part, the aggregation..." then it will read correctly.

7. The authors should probably note somewhere that the recall data involves somewhat of an arbitrary choice about when no retrieval was occurring. Other paradigms involving forced choices may be better suited to address the timing of incorrect responses. This is a minor detail though and simply needs some mention in the discussion.

8. Discussion: "fMRI connectivity literature also demonstrates..."

Many of these citations are out of date or involve only limited nodes. A more up to date literature, involving a much more extensive use of nodes, showing similar findings to here, would involve studies by King et al. 2015 J Neuro, Schedlbauer et al. 2013 Scientific Reports, and Geib et al. 2015 Cerebral Cortex.

Reviewer #2 (Remarks to the Author):

This study uses a dataset of intracranial EEG (iEEG) recordings from 300 patients that have been recorded in a memory paradigm. The authors analyze power and synchronization in two pre-specified frequency bands referred to as gamma and theta. For different memory-related contrasts, they report primarily gamma desynchronization and theta synchronization. Furthermore, they find gamma desynchronization to be accompanied by

gamma power enhancements.

One major problem of this study is that it is mainly incremental in comparison to a previous paper from the same group: J Neurosci. 2013 Jan 2;33(1):292-304. doi: 10.1523/JNEUROSCI.2057-12.2013. Synchronous and asynchronous theta and gamma activity during episodic memory formation. Burke JF1, Zaghoul KA, Jacobs J, Williams RB, Sperling MR, Sharan AD, Kahana MJ. As discussed in lines 248 – 253 of the present manuscript, one of the main advances of the present study is to move from lobe-wise interactions to ROI-wise interactions. This is an incremental advance.

The study does not show the presence of the investigated rhythms, neither of theta nor of gamma. Without convincing evidence for a theta rhythm and a gamma rhythm, the authors cannot claim that they investigate theta or gamma. Convincing evidence could e.g. be clear peaks in power spectra or phase-locking spectra.

It is a major weakness that the authors ignore other frequency bands. Had they first investigated power or phase-locking spectra, they would have most likely found alpha and beta to be present in their data. Those rhythms have also been linked to memory. Why were they omitted?

Several previous studies have shown that iEEG signals in the gamma-frequency range often reflect primarily action potentials (APs) and/or postsynaptic potentials (PSPs) with broadband power signatures, rather than a genuine gamma rhythm. The authors appear fully aware of this. However, they fail to see the consequences of this. The broadband high-frequency power should not be referred to as gamma, because it is not a rhythm, like the alpha, beta and gamma rhythms are. The results do not support the conclusions, because they rely on the assumption that the analysis is concerned with intracranially recorded EEG rhythms. Rather, the analysis actually deals with signals that partly reflect the largely asynchronous local APs and/or PSPs. It is known that APs/PSPs show much weaker long-range phase-locking than EEG rhythms. A difficult challenge for robust conclusions from the present analysis is the fact that the recorded signals reflect mixtures of EEG rhythms and broadband AP/PSP reflections. One plausible interpretation of the presented results is the following: The power in a given frequency band is a mixture of a genuine rhythm and the broadband power. In the analyzed theta-frequency range, this mixture might contain relatively more rhythmic components, in the analyzed gamma-frequency range, the mixture might contain more broadband components. The rhythmic components actually show some degree of long distance synchronization. When a brain area is activated, the broadband power increases and thereby its relative contribution to the recorded mixed signal. As the APs/PSPs leading to the broadband power are largely not long-distance synchronized, the increase of their contribution to the signal leads to a reduction in the long distance phase locking.

The main contribution of the paper is to document differences in long distance phase locking in several memory-related contrasts, and then to further characterize the network properties of this. The presentation of these results is cast in the context of the theta and gamma rhythms. However, the problems arising through signal mixing prohibit these

interpretations. Unfortunately, it is also not a solution to simply change the wording and replace e.g. „gamma“ by „broadband“. The analysis is fundamentally confounded by the fact that the signal mixing, i.e. the relative contributions of rhythmic and broadband components, changes between conditions. And the interpretation depends on the analysis dealing with real rhythms, because changes in phase locking upon changes in signal mixing do not provide insights into physiology, but merely reflect artifacts of signal mixing.

The authors actually seem to fully agree to this interpretation, as they write e.g. in lines 60-64: „If gamma activity is not synchronous, it may instead reflect an aggregation of rapid, stochastic firing in a population of neurons near an electrode, not an oscillatory modulation of activity that indicates coordinated activity across space. Were this true, the general neural activation of a brain region – captured by the spectral power recorded at a cortical electrode – would rise as the synchronicity of that region with others tends to fall.“ The problem is that with this interpretation, the paper does not reflect any significant scientific advance, certainly not one that should be reported in Nature Communications.

It is often not clear, whether the authors refer to synchronous activity itself or memory related changes thereof. For example, line 100 states that the network-wide level of synchronous activity in gamma was not significant. This sounds like reporting the level of synchronous activity independent of a memory contrast. If so, how was significance tested? Another example is in lines 114 – 118, where the authors refer to most highly connected ROIs. Also, they often refer to connections as „asynchronous“ or „synchronous“, while meaning „decreasing in synchronous activity in a memory contrast“ or „increasing in synchronous activity in a memory contrast“. They need to revise this throughout the manuscript to properly refer to the respective memory contrasts, wherever this applies.

Line 172: It is not fully clear how the power-synchrony correlation in Fig. 4A is calculated. Is this a correlation across time-frequency pixels?

Line 202: The authors write that gamma power and theta synchrony were positively correlated, but Fig. 4B fails to show that.

To which degree do the reported synchronization phenomena, and their differences between memory conditions, reflect stimulus locking? This is particularly relevant for theta.

Lines 95-96 clarify that permutations were between remembered and not-remembered trial labels. This statement should also be clearly made in line 644, which one could also misunderstand as an independent shuffling of trials between the signals for which the phase locking is determined.

Did the authors control for eye fixation? Can differences in eye position and corresponding differences in visual input explain some of the differences ascribed to memory?

I presume that words were presented visually. But is this actually stated explicitly?

The dataset with 300 subjects is impressive. But why then do the authors not use this to

perform random effect analyses? The trial shuffling described in the methods probably implements merely a fixed effect analysis.

Lines 615 – 624: The resultant vector length shows a bias that grows with small sample size and with small true phase locking value. The authors need to correct for this, at least if sample sizes differ between compared memory conditions. This corrections seems to be completely missing.

How was the position of a bipolar electrode pair assigned to a ROI? Do the authors exclude bipolars that crossed ROI boundaries?

Line 697: Why is alpha set to 0.1 here?

Lines 700 – 710: The authors perform some selections here, that appear arbitrary. How can those selections be justified? Can they be avoided?

Lines 732-735: This seems to be not a shuffle, but a lagged correlation. If there is a side trough, this might give a false positive test.

Reviewer #3 (Remarks to the Author):

Review of Solomon et al.

Solomon et al. study connectivity and power dynamics in theta and gamma bands in ECoG signals recorded from a large dataset of 300 subjects during a memory encoding and retrieval task. They show that in the gamma band, phase differences between two areas are less clustered for words that are later remembered versus words that are not remembered, implying that memory encoding leads to desynchronization across areas in the gamma band. In the theta band, the opposite trend is observed. Further, gamma power is negatively correlated with gamma synchronization across brain areas, but positively correlated with theta synchronization when the analysis is restricted to memory encoding areas.

Overall, I think the results are interesting, data size is very impressive, and the analysis is thorough. Theta-gamma coupling has invoked a lot of interest in recent years, and this work substantially adds to this body of work. I have some questions and clarifications, as discussed below, but they should not be too difficult to address.

Major points

1. Convention: In a lot of previous studies, "gamma band" typically refers to the "bandlimited" gamma, which is typically between 30-70 Hz, as opposed to the "broadband" gamma, which is at a higher frequency and is often called high-gamma to distinguish it from the lower frequency gamma. The last author of this manuscript has a very influential paper

himself, where they linked broadband gamma to spiking activity (Manning et al., 2009). Although the authors mention this in the discussion, I feel the distinction could be made clearer that we're talking about the broadband gamma (or high-gamma) here, not the band-limited gamma. The analysis performed between 45-95 Hz could also be done in the typical gamma range, say between 30-60 Hz, and a higher frequency range, say between 100-150 Hz, to test whether the results are valid only between 45-95 or extend to other frequencies as well.

2. I had difficulty understanding some aspects of the analysis. My understanding is as follows: each trial has a sequence of 12 words, followed by delay and retrieval. During retrieval, some of those 12 words are remembered and others not. Each experiment has up to 25 trials (up to 300 words). Out of this full set of ~300 words, the phase difference distribution is computed for remembered and non-remembered words separately to get vectors R1 and R2, from which the F-statistic is computed. Is this correct? If so, then when the author's say "remembered trials" (say at line 613), they mean "remembered words", right? Or am I missing something?

3. Related to the previous point, the number of vectors used for obtaining R1 and R2 is critical, but it is not described very clearly. It would be nice if some statistics are provided regarding the percentage of correct recalls in the population, as well as the number of trials performed (mean +- SEM), so that we know the number of trials of each condition (n1 and n2 in the F statistic formula).

4. The resultant vector length (R) critically depends on the number of trials. It is unclear how the multipliers used in the F-statistic equation resolve this issue. Some pointers should be provided why the F-statistic was chosen the way it was. Because R1 and R2 are already normalized between 0 and 1, I do not understand why the vector lengths are divided, because large vector lengths in R1 would lead to potentially very large values of the F-statistic. For example, why are the vector lengths not subtracted instead (after correcting for the bias due to unequal number of trials)? I do believe that the randomization test takes care of most of these issues (including potential biases due to unequal number of trials), but some pointers as to why the statistic was chosen in this particular way would be very useful.

5. Line 692: "we sum the number of significant time-frequency points". What is being summed? The total number of significant points, or the unthresholded z-scores as mentioned on Line 115?

6. How important is bipolar referencing for these results? It could be the case that gamma synchronizes over smaller regions, but bipolar subtraction takes out the common component. The differences shown here could be just due to different space constants associated with gamma and theta networks. I understand that re-doing the analysis for the entire dataset could be difficult, but can the authors compare the results for unipolar versus bipolar referencing for at least a partial dataset to test whether the results hold across referencing techniques?

7. The evoked response (average of the ECoG time series traces) often has power at low frequencies, especially in the theta range. Therefore, regions that produce an evoked response might show a corresponding theta-band power/synchrony. It would be interesting to see whether the results hold for the induced theta and gamma as well (obtained by first subtracting the evoked response from each trace). This could also be done on a subset of data if re-doing the analysis on 300 subjects is difficult.

Summary of Major Responses

Before enumerating each reviewer's comments line-by-line on the next page, we first include a high-level summary of the major revisions we made to the manuscript based on the reviewers' feedback:

1. Reviewer 2 asked us to better articulate how our work goes significantly beyond previous studies. As our revised Discussion makes clear, this manuscript reports three major advances: First, we establish for the first time that decreases in synchrony accompany increases in high-frequency power, and that this fundamental relationship between power and synchrony manifests itself throughout the human brain. Second, whereas previous reports of desynchronization between broad brain regions left open the possibility of synchrony at a finer spatial scale, our analysis of 30,000 electrodes allowed us to establish that nearly all subregions of the memory network exhibit asynchronous activity during successful memory encoding. Third, we demonstrate that the same spectral signature that characterizes successful memory encoding (asynchronous high-frequency and synchronous low-frequency activity) also appears just prior to the spontaneous recall of items. This finding shows that the encoding-related desynchronization of high-frequency activity cannot be driven by the subject's response to an external stimulus.
2. Reviewer 2 noted that our original manuscript did not differentiate oscillatory and non-oscillatory sources of high-frequency activity. To address this potential confound, we embarked on a substantial new analysis that decomposes these two sources of high-frequency activity (Fig. 7 in the main text, Fig. 2 in the response letter). Doing so allowed us to isolate a small number of electrodes that exhibited clear memory-related oscillations. At these electrodes, as the reviewer hypothesized, we find evidence of enhanced high-frequency synchronization during memory encoding. The absence of this pattern in the overall dataset reflects the minority contribution of oscillatory signals towards memory processing (Fig. 7D). These additions strengthen our manuscript in that (1) we quantify the extent to which human memory is correlated with oscillatory versus broadband changes in spectral activity, and (2) we show that high-frequency oscillatory activity is associated with long-range synchronization.
3. Reviewers 2 and 3 asked why we did not extend our analyses to other frequencies implicated in human memory. We now include an analysis of frequencies from 3-120 Hz (Supplementary Figures S2, S5), revealing asynchronous activity in 30+ Hz bands and synchronous activity from 3-28 Hz. We found particularly strong synchronous activity associated with good memory in the alpha (9-13 Hz range). Furthermore, we observed a strong inverse correlation between spectral power and synchrony across these frequency bands (Supplementary Figure S5). This finding aligns with our initial reports of the relationship between spectral power and synchrony in only the high gamma and theta bands.
4. All reviewers had general questions about our methods, and asked that some analyses be justified more clearly. In response to this feedback, we have substantially simplified several of our methods. Principally, in response to Reviewer 3's suggestion, we now simply take the difference of resultant vector length values as our primary metric of memory-related synchronization/desynchronization. We more clearly justify the use of a nonparametric permutation test to account for the sample size bias inherent in these statistics. In response to Reviewers 1 and 2, we also present a random-effects analysis in the response letter (Fig. 4), and demonstrate a strong correlation with our fixed-effects analysis. Finally, the revised manuscript draws on a larger dataset than the original; though some subjects were now excluded due to sampling rates too low to assess our upper frequencies, additional subjects with more electrodes were introduced. Accordingly, we now draw on a dataset with approximately 3,000 more electrodes than previously.

Responses to Reviewer Comments

We address each reviewer's comments below. All references to figures refer to those included in this letter, unless otherwise noted.

Reviewer #1 (Remarks to the Author):

Overall assessment: This is an extremely important and well-considered study that will have a significant impact on the field. This is true for many reasons. For one, the study is the first to directly test the "gamma vs. theta" synchrony hypothesis, decidedly resolving this debate clearly in the favor of theta. This helps resolve decades of debate on this topic. Another important component of the study is the identification not only of hubs that in turn connect to the memory literature (i.e., MTL), but their specific time courses. Additionally, the study nicely ties together disparate work that has typically treated encoding and retrieval separately and links them under the same rubric of network coherence. Finally, the study is high-powered and the experimental paradigm both robust yet ecologically valid and helpful. I have only minor concerns aimed at improving the breadth and impact of the findings, which I detail below.

- 1. Methods: The authors indicate that they use bipolar montages. This makes sense for analysis of local power but it is less clear how this is optimal for looking at global synchrony. In fact, many past papers testing such issues have used an averaged reference. Do the results generally hold if the authors use an averaged reference, based on areas that do not include ictal or inter-ictal activity?*

We thank the reviewer for raising this question. We had initially considered whether to use an average montage or a bipolar reference scheme. Each of these methods has unique strengths and weaknesses. Indeed, the average reference -- reflecting inconsistent and often sparse electrode coverage across subjects -- may also introduce distortions in phase synchrony or coherence estimates (Nunez et al. 1997; Schiff 2005; Guevara et al. 2005).

We therefore were left with two flawed options for a montage, but further had to consider a critical component of our analysis: a comparison of phase synchrony to local spectral power. Because we wanted to assess high-frequency activity with high spatial precision -- as this is critical for comparing activity in neighboring brain regions -- we followed several previous papers in using a bipolar reference scheme (e.g. Burke et al. 2013; Merkow et al. 2014; Ezzyat et al. 2017, among others). Bipolar referencing is arguably better suited for the analysis of gamma and high-frequency activity than for the analysis of low frequency activity, especially in the delta and theta bands, as this activity tends to be more spatially autocorrelated. Nonetheless, the relatively wide spacing of electrodes used in these studies (typically 1 cm) enables the evaluation of theta activity with bipolar referencing (see Raghavachari et al. 2001).

In our study, we also excluded connectivity values computed between pairs of bipolar electrodes if that pair shared a monopolar contact, accounting for the possibility that a shared signal may contaminate connectivity between a pair of virtual electrodes. We strongly agree that future work should seek to quantify how montage choice affects measures of inter-regional synchronization.

- 2. Methods: The use of "F statistic" is unclear. What is this referring to? Why were they summed?*

The F-statistic, available through the Circular Statistics Toolbox ("ktest"), captures the ratio of the concentrations of two phase difference distributions. Here, it was used in a non-parametric fashion as part of an analysis to construct whole-brain connectivity maps, wherein the F-statistics and their corresponding null distributions were averaged across electrodes and summed across subjects to derive an overall synchronization effect per ROI pair.

We have simplified our methods in the revised manuscript. We now simply take the difference of the resultant vector lengths between the remembered and not-remembered phase difference distributions, and compare that

difference to the distribution expected under chance (shuffled trial labels). The non-parametric shuffling procedure used to create a distribution of differences expected by chance account for biases in resultant vector length values computed from different sample sizes (since the number of vectors in each group, “remembered” and “not-remembered,” is not changed when merely trial labels are shuffled). Like before, the true differences and null differences are averaged across electrodes that span a pair of ROIs within a subject, and then the resulting statistics are summed across whichever subjects have electrodes spanning that pair of ROIs, to derive an average synchronization effect per ROI pair.

3. *The N varies widely for different electrode placements. The extensive coverage is a major strength of the paper. Yet, the inconsistent placement across subjects, ranging from 7-200 subjects overlapping with the same area, is a potential weakness. Can the authors perform an analysis in which they (roughly) equate subject number, perhaps randomly choosing subjects so N=20, to show that the results still hold?*

It is true that different ROI pairs have synchronization values that may draw on very different amounts of data. Here we show an analysis of networks with ROIs of approximately equated subject number ($15 < N < 25$), as suggested (Fig. 1). Results are consistent with the findings among the full dataset.

Figure 1. ROI-pairs with approximately equal sampling show consistent results with full dataset. *Top row:* Theta and high gamma (HG) adjacency matrices representing the change in phase synchronization in remembered vs. not-remembered conditions. Only ROI pairs with between 15 and 25 subjects were included (gray pixels did not meet criteria). *Bottom row:* The overall level of synchronous connection in theta, and the overall level of asynchronous connections in HG (45-100 Hz), are significantly above chance ($P < 0.01$), consistent with findings in the whole dataset (see main text, Figure 1).

We also believe the reviewer’s concern is more generally addressed by the random effects analysis we present in Fig. 4 here. We demonstrate that pairwise connectivity effects observed in aggregate also generalize to the clinical population we sample from, as captured by a random effects inference on each ROI pair.

4. *In many places (Figure 1F, Figure 6F), the authors imply an interaction effect but do not directly test this. Can they use a Chi Square test of association to explicitly test for a cross-over interaction effect based on edge weight?*

We now report the chi-square test for these data in the figure captions, which indicates a significant synchrony-frequency interaction.

5. *Abstract: the authors state “Yet, the absence of high-resolution data on whole brain electrical connectivity during cognitive tasks has prevented a direct test of this hypothesis.” This statement needs amending. In fact, numerous other intracranial studies have used multilobular grids and depth electrodes to look at memory, and cognition more generally. I think what the authors are trying to argue is that their coverage, because of their large sample size, is more extensive. This sentence should be rephrased.*

We thank the reviewer for noting this confusing phrasing. We have amended the sentence to now read: “Yet, insufficient data on intracranial electrical connectivity has precluded a direct test of this idea in a whole-brain setting.”

6. *p. 11: “This relation is consistent with the hypothesis that broadband high-frequency...”*

I think the authors intended to state that gamma is a reflection, in part, of spiking activity as well as synaptic activity. If they add “in part” to the phrasing of the sentence “reflects, in part, the aggregation...” then it will read correctly.

We agree -- adding “in part” captures our intended meaning, and we believe that broader changes in the discussion section, in response to comments from reviewers, better reflect this idea.

7. *The authors should probably note somewhere that the recall data involves somewhat of an arbitrary choice about when no retrieval was occurring. Other paradigms involving forced choices may be better suited to address the timing of incorrect responses. This is a minor detail though and simply needs some mention in the discussion.*

Though our focus in this manuscript was the large dataset collected on a free-recall task, we agree that recognition tasks would provide more precise timing in a retrieval analysis -- this is now mentioned in the Discussion on page 14. However, we also note that others who have used this retrieval contrast show highly reliable univariate and cross-validated multivariate correlations with memory states, suggesting its utility for analyses in this domain (Burke et al. 2014; Kragel et al. 2017).

8. *Discussion: “fMRI connectivity literature also demonstrates...”*

Many of these citations are out of date or involve only limited nodes. A more up to date literature, involving a much more extensive use of nodes, showing similar findings to here, would involve studies by King et al. 2015 J Neuro, Schedlbauer et al. 2013 Scientific Reports, and Geib et al. 2015 Cerebral Cortex.

We thank the reviewer for these helpful citations, and we now make reference to more relevant and timely works. These references align well with our graph-theoretic approach and our finding that large-scale, diffuse low-frequency networks underlie successful memory processing.

Reviewer #2 (Remarks to the Author):

1. *One major problem of this study is that it is mainly incremental in comparison to a previous paper from the same group: J Neurosci. 2013 Jan 2;33(1):292-304. doi: 10.1523/JNEUROSCI.2057-12.2013. Synchronous and asynchronous theta and gamma activity during episodic memory formation. Burke JF1, Zaghloul KA, Jacobs J, Williams RB, Sperling MR, Sharan AD, Kahana MJ. As discussed in lines*

248 – 253 of the present manuscript, one of the main advances of the present study is to move from lobe-wise interactions to ROI-wise interactions. This is an incremental advance.

In the revised manuscript, we now more thoroughly describe how the present work goes substantially beyond prior work, including Burke et al (2013). To our knowledge, no prior study (including Burke et al., 2013) has examined the relation between local power and inter-regional synchronization across the whole brain. Not only does our study address power/synchrony relationships in a general, brain-wide sense (Fig. 5, main text), we quantify how these relationships are variable across regions and frequency bands (Fig. 4A, main text). In doing so, we firmly establish an inverse relationship between local high-gamma (HG) and inter-regional HG connectivity -- widely believed, to be sure, but a belief that deserves scientific proof. Our study is that proof.

By aggregating entire lobes in their analysis of asynchronous high frequency activity, Burke, et al. (2013) left open the possibility that HG synchronization takes place on finer anatomic scales, as suggested by work from other groups (Axmacher et al. 2006). Therefore, that study was not sufficiently designed to answer whether HG synchronization correlates with successful memory encoding. Moreover, that study did not extend its results to a retrieval period, limiting the strength of its claims about general brain function.

Finally, in response to this reviewer's comments, we have incorporated a new analysis that takes this manuscript even further beyond the Burke, et al. (2013) study. In the response to the next question, we address this new analysis in more detail, though briefly: After quantifying whether each electrode exhibits a change in oscillatory high-frequency power, we reconstruct our brain-wide network with those electrodes only. We show that, as this reviewer hypothesized, oscillatory activity appears to underlie high-frequency synchronization. However, the occurrence of these oscillations is extremely rare (see Fig. 2).

2. *The study does not show the presence of the investigated rhythms, neither of theta nor of gamma. Without convincing evidence for a theta rhythm and a gamma rhythm, the authors cannot claim that they investigate theta or gamma. Convincing evidence could e.g. be clear peaks in power spectra or phase-locking spectra.*

We thank the reviewer for making this excellent point. We also believe it is important to quantify the degree to which true rhythms are present in our data. Accordingly, we have embarked on a substantial new analysis for this revised manuscript.

We adopt a widely-used method for oscillation detection, called P_{episode} or BOSC (van Vugt et al. 2007; Whitten et al. 2011; Hughes et al. 2012). This method applies two criteria for the identification of a true oscillation: a minimum time (at least 3 cycles), and a significant deviation of spectral power from an expected linear fit to the log-power vs. log-frequency curve. For each electrode in our dataset, we used BOSC to ask whether the presence of low gamma (30-58 Hz) oscillations were correlated with whether a word would later be remembered or forgotten. We found that a small fraction of electrodes in our dataset exhibited reliable, memory-related oscillatory gamma power (approximately 260). Far more electrodes exhibited non-oscillatory (i.e. broadband or mixed) subsequent memory effects in both the 30-60 Hz and the 65+ Hz range (Fig. 2D).

Next, among only that subset of electrodes, we computed phase synchronization at the specific epoch and frequency where an oscillation was observed. We found that, indeed, positive memory-related gamma phase-synchrony was observed among this restricted subset of electrodes (Fig. 2E, 2F). Accordingly, we have directly quantified the degree to which high-frequency memory networks reflect broadband asynchrony and narrow-band oscillatory synchronization. The overwhelmingly majority of electrodes feature broadband asynchrony (Fig. 2D), but where oscillations are present, so is inter-regional synchronization.

Figure 2. Synchronization of gamma (30-60 Hz) oscillatory activity. (A) Example of an electrode exhibiting a gamma oscillation in the middle occipital gyrus, as detected by BOSOC (see Methods for details). Red line reflects average log power across all remembered events, blue line reflects average log power across not-remembered events. An isolated peak in the power spectrum is indicated, between approximately 36 and 74 Hz. (B) For the electrode in (A), heatmap of the t-statistic reflecting the relative frequency of oscillations detected in remembered versus not-remembered trials. Red colors indicate more oscillations detected at given frequency and timepoint in trials that were later remembered correctly. Increased oscillatory power from approximately 43 Hz to 58 Hz, coincident with the oscillatory peak in (A), is indicated. (C) Count of all electrodes in the 294-subject dataset that exhibit an oscillatory subsequent memory effect (SME) between 30 and 58 Hz, at each 200ms epoch spanning the word presentation interval (see Methods for details). The greatest number of electrodes exhibit oscillatory SMEs at 52 Hz, between 400 and 600 ms after onset of a word (black box). (D) Count of electrodes in the dataset that exhibit different kinds of SMEs between 400 and 600 ms: 52 Hz oscillation, 30-60 Hz power, or 65-100 Hz power. (E) Count of significant pairwise-ROI synchrony/asynchrony among the subset of electrodes exhibiting 52 Hz oscillatory SME at 400-600ms. Left: Counts observed at 50-55 Hz, near the frequency of maximal oscillatory SMEs (52 Hz). Right: Counts observed in the 65-85 Hz range among the same electrode subset. (F) Average network synchrony (z-score) for the subnetwork of regions sampled in the 52 Hz oscillatory electrode subset, measured by summing the subnetwork connection weights at each frequency in the 400-600 ms window.

3. *It is a major weakness that the authors ignore other frequency bands. Had they first investigated power or phase-locking spectra, they would have most likely found alpha and beta to be present in their data. Those rhythms have also been linked to memory. Why were they omitted?*

We agree that this analysis should be broadened to additional frequency bands. In the revised manuscript, we now compute phase-synchrony networks in the alpha (9-13 Hz) and beta (16-28 Hz) ranges, and extend our

gamma range to a minimum of 30 Hz and a maximum of 120 Hz. We observe significant memory-related synchrony in the theta (3-8 Hz), alpha, and beta ranges, and significant memory-related asynchrony in the low gamma (30-60 Hz), 65-85 Hz, and 90-120 Hz ranges (Fig. 3). We note particularly strong effects in the alpha range, which warrants further investigation in future studies. Additionally, in Supplementary Figure S5, we show how the inverse power-synchrony relationship is maintained across these new frequency bands.

Figure 3. Synchrony effects from 3-120 Hz. *A, left:* Brainwide phase synchronization SME in the memory encoding interval. Red reflects increased synchrony associated with successful memory encoding, blue reflects decreased synchrony. Z-scores of less than 2 are faded, vertical black lines indicate word onset and offset. *Right:* Brainwide spectral power in SME in the encoding interval. *B:* Overall level of synchrony/asynchrony in each frequency band during the word presentation interval.

- Several previous studies have shown that iEEG signals in the gamma-frequency range often reflect primarily action potentials (APs) and/or postsynaptic potentials (PSPs) with broadband power signatures, rather than a genuine gamma rhythm. The authors appear fully aware of this. However, they fail to see the consequences of this. The broadband high-frequency power should not be referred

to as gamma, because it is not a rhythm, like the alpha, beta and gamma rhythms are. The results do not support the conclusions, because they rely on the assumption that the analysis is concerned with intracranially recorded EEG rhythms. Rather, the analysis actually deals with signals that partly reflect the largely asynchronous local APs and/or PSPs. It is known that APs/PSPs show much weaker long-range phase-locking than EEG rhythms. A difficult challenge for robust conclusions from the present analysis is the fact that the recorded signals reflect mixtures of EEG rhythms and broadband AP/PSP reflections. One plausible interpretation of the presented results is the following: The power in a given frequency band is a mixture of a genuine rhythm and the broadband power. In the analyzed theta-frequency range, this mixture might contain relatively more rhythmic components, in the analyzed gamma-frequency range, the mixture might contain more broadband components. The rhythmic components actually show some degree of long distance synchronization. When a brain area is activated, the broadband power increases and thereby its relative contribution to the recorded mixed signal. As the APs/PSPs leading to the broadband power are largely not long-distance synchronized, the increase of their contribution to the signal leads to a reduction in the long distance phase locking.

The main contribution of the paper is to document differences in long distance phase locking in several memory-related contrasts, and then to further characterize the network properties of this. The presentation of these results is cast in the context of the theta and gamma rhythms. However, the problems arising through signal mixing prohibit these interpretations. Unfortunately, it is also not a solution to simply change the wording and replace e.g. „gamma“ by „broadband“. The analysis is fundamentally confounded by the fact that the signal mixing, i.e. the relative contributions of rhythmic and broadband components, changes between conditions. And the interpretation depends on the analysis dealing with real rhythms, because changes in phase locking upon changes in signal mixing do not provide insights into physiology, but merely reflect artifacts of signal mixing.

The authors actually seem to fully agree to this interpretation, as they write e.g. in lines 60-64: „If gamma activity is not synchronous, it may instead reflect an aggregation of rapid, stochastic firing in a population of neurons near an electrode, not an oscillatory modulation of activity that indicates coordinated activity across space. Were this true, the general neural activation of a brain region – captured by the spectral power recorded at a cortical electrode – would rise as the synchronicity of that region with others tends to fall.“ The problem is that with this interpretation, the paper does not reflect any significant scientific advance, certainly not one that should be reported in Nature Communications.

Related to the above questions, we have sought to quantify the degree of signal mixing in our results with an additional analysis that identifies oscillations (Fig. 2). Though we stand by our original results as a significant advance, detailed in response to this reviewer’s first question, we agree that the results could be strengthened further by separately analyzing oscillatory and non-oscillatory components. We have done so in the revised manuscript and describe those results on pages 5 and 6 of this letter. To reiterate, the number of electrodes exhibiting a memory-related change in oscillatory gamma power is small (approx. 260), and even that number arises from a liberal significance criteria (see Methods). This is an extremely small fraction of the 27,000 electrodes in the dataset (approx. 1%). Far more electrodes exhibit an increase in spectral power without assessing for oscillations in the 30-60 Hz band (approx. 1400), and even more show power SMEs at higher frequencies (65-100 Hz, approx. 2500 electrodes). However, among the 260 electrodes that exhibit an oscillatory SME, a positive phase synchrony SME is observed (Fig. 2).

5. *It is often not clear, whether the authors refer to synchronous activity itself or memory related changes thereof. For example, line 100 states that the network-wide level of synchronous activity in gamma was not significant. This sounds like reporting the level of synchronous activity independent of a memory contrast. If so, how was significance tested? Another example is in lines 114 – 118, where the authors refer to most highly connected ROIs. Also, they often refer to connections as „asynchronous“ or „synchronous“, while meaning „decreasing in synchronous activity in a memory contrast“ or „increasing*

in synchronous activity in a memory contrast“. They need to revise this throughout the manuscript to properly refer to the respective memory contrasts, wherever this applies.

We thank the author for noting these discrepancies. The entire manuscript analyzes a *difference* in synchronicity between “remembered” and “not-remembered” states (or the analog for our retrieval condition). At several places we have revised our language to better emphasize this contrast (e.g. “correlated with successful memory encoding/retrieval”).

6. *Line 172: It is not fully clear how the power-synchrony correlation in Fig. 4A is calculated. Is this a correlation across time-frequency pixels?*

The reviewer is correct. We have made a minor modification to the revised manuscript’s methods for clarity.

7. *Line 202: The authors write that gamma power and theta synchrony were positively correlated, but Fig. 4B fails to show that.*

At this line, we were referring to the relationship between gamma power and theta synchrony in the core memory network (Fig. 5). The relevant line now reads as follows:

“In the core memory network, we found that gamma-power and gamma-synchrony dynamics were inversely related, while gamma-power and theta-synchrony were positively correlated.”

8. *To which degree do the reported synchronization phenomena, and their differences between memory conditions, reflect stimulus locking? This is particularly relevant for theta.*

The reviewer raises a good question. We interpret it as follows: Is it possible that stimulus presentation resets the phase of ongoing oscillations, and this reset occurs more often in trials that were later remembered vs. forgotten? If so, could our observed low-frequency phase synchrony arise due to these evoked potentials and phase resetting, rather than induced low-frequency oscillations? There are several reasons we believe stimulus-locking does not significantly influence our findings.

- a) As seen in Fig. 3 in the main text and Fig. 3A in this letter (Fig. S2 in Supplementary Figures), strong synchrony effects are observed in the theta/alpha range as late as 1200-1600 ms after onset of the stimulus. We have quantified the late-phase theta synchrony effect below, and it is highly significant ($P < 0.01$). Evoked responses would be unlikely to influence dynamics so long after stimulus presentation; we interpret these findings to mean induced activity drives the effect. More generally, maximal theta/alpha synchrony occurs in the 600-1000ms range, beyond the timescale of typical ERPs.

- b) Our results generalize to the retrieval period, in which there is no stimulus at all. (We assess an epoch starting 500 ms prior to vocalization of a word as contrasted to extended periods of time in which the subject produces no vocal responses, see Methods for details.)
 - c) A strong low-frequency stimulus locking effect, preferentially present in the remembered vs. not-remembered condition, would manifest as increased low-frequency spectral power at those frequencies. In fact, consistent with prior literature, we find a widespread decrease in low-frequency power shortly after word onset (see Fig. 3A).
9. *Lines 95-96 clarify that permutations were between remembered and not-remembered trial labels. This statement should also be clearly made in line 644, which one could also misunderstand as an independent shuffling of trials between the signals for which the phase locking is determined.*

In the initial manuscript, “trial labels” was the language used on line 644 and 95-96.

10. *Did the authors control for eye fixation? Can differences in eye position and corresponding differences in visual input explain some of the differences ascribed to memory?*

We do not explicitly control for eye fixation. Issues related to stimulus locking, however, are addressed above. It would also be difficult to understand how eye position corresponds to differences observed in the retrieval contrast, when subjects are asked to freely recall words from the encoding interval.

11. *I presume that words were presented visually. But is this actually stated explicitly?*

This was not stated explicitly; the methods section has been updated accordingly.

12. *The dataset with 300 subjects is impressive. But why then do the authors not use this to perform random effect analyses? The trial shuffling described in the methods probably implements merely a fixed effect analysis.*

The reviewer raises a good point – the fixed effects analysis performed here does not answer the question as to whether the observed synchrony dynamics are typical of the population from which we sample. To better answer this question, we performed a random effects analysis: A z-score for the memory-related change in synchronization is computed per-subject, per-ROI pair. For any ROI pair with more than 10 subjects, a one-sample T-test is used to test whether the mean of the z-score distribution significantly deviates from zero.

As shown below, our fixed effects and random effects analyses are highly consistent, as captured by the correlation of the HG adjacency matrices during the encoding interval (Fig. 4). Assessing hub scores (and subsequently, power-synchrony relationships) across subjects is a more difficult task, since differential electrode coverage will strongly influence an ROI's node strength. In other words, the only way to assess a ROI's node strength in a whole-brain setting is to use data aggregated across individuals in our dataset. Future work should seek to identify subjects that have roughly similar coverage, such that node strength can be reliably compared across individuals.

Figure 4: Random versus fixed-effects analysis. A: Adjacency matrix reflecting random-effects phase synchrony SMEs. Each pixel represents the one-sample t-statistic comparing subject Z-scores against zero (for all ROI pairs with 10 or more subjects). B: Example analysis of one ROI pair (left postcentral gyrus vs. left middle temporal gyrus). Top: Average z-score over time across 97 subjects, error bars ± 1 SEM. Bottom: Histogram of subject z-scores for this ROI pair during the encoding interval. C: Correlation of random effects t-statistics and fixed effects z-scores for all ROI pairs, during the encoding interval.

13. Lines 615 – 624: The resultant vector length shows a bias that grows with small sample size and with small true phase locking value. The authors need to correct for this, at least if sample sizes differ between compared memory conditions. This corrections seems to be completely missing.

It is for this reason that we use a non-parametric permutation test for significance. In the null condition of shuffled trial labels, the number of trials in each surrogate group are the same as in the true groups. Therefore, the distribution of possible resultant vector length differences under chance reflects this bias as well -- true differences are compared to this null to generate a z-score or p-value. In other words, the null distribution for the synchronization effect at each electrode may not be centered around zero, if the number of trials in each condition are sufficiently different. For clarity, we have made this idea explicit in the methods section.

14. How was the position of a bipolar electrode pair assigned to a ROI? Do the authors exclude bipolars that crossed ROI boundaries?

As described in the methods section, the position of a “virtual” bipolar electrode is given by the midpoint of the two monopolar electrodes. We use this midpoint to determine ROI assignment. We do not exclude bipolar that cross ROI boundaries, though we agree that such a correction would be valuable for future studies.

15. *Line 697: Why is alpha set to 0.1 here?*

We agree that this choice was arbitrary. The revised manuscript now only reports hubs with a corrected p-value of less than 0.05, and further indicates which hubs are significant with $P < 0.01$ (Fig. 2 in main text).

16. *Lines 700 – 710: The authors perform some selections here, that appear arbitrary. How can those selections be justified? Can they be avoided?*

In lines 700-702, the intent was to achieve more interpretable brain plots (Fig. 2 in main text) by limiting the number of displayed connections (by only choosing the 5 strongest, so long as the connection strength z-score exceed 2.5). These steps are taken for visualization only; connections are not thresholded for hub analyses or synchrony-power analyses.

In lines 703-710, regarding the plots of hub timecourses (Fig. 3 in main text), we selected these regions to display on the basis of where hubs were observed and a focus on popular regions in the memory literature. However, we have included additional timecourse data in the supplementary figures.

17. *Lines 732-735: This seems to be not a shuffle, but a lagged correlation. If there is a side trough, this might give a false positive test.*

We believe this test is more conservative than a shuffle. But not disturbing the autocorrelated spatiotemporal structure of either the power values or synchrony values, we account for the non-independence of each datapoint and create a strong null. For a true correlation to be significant, it must exceed correlations that arise even from small shifts of these spatiotemporal vectors against one another.

Reviewer #3 (Remarks to the Author):

1. *Convention: In a lot of previous studies, “gamma band” typically refers to the “bandlimited” gamma, which is typically between 30-70 Hz, as opposed to the “broadband” gamma, which is at a higher frequency and is often called high-gamma to distinguish it from the lower frequency gamma. The last author of this manuscript has a very influential paper himself, where they linked broadband gamma to spiking activity (Manning et al., 2009). Although the authors mention this in the discussion, I feel the distinction could be made clearer that we’re talking about the broadband gamma (or high-gamma) here, not the band-limited gamma. The analysis performed between 45-95 Hz could also be done in the typical gamma range, say between 30-60 Hz, and a higher frequency range, say between 100-150 Hz, to test whether the results are valid only between 45-95 or extend to other frequencies as well.*

We thank the reviewer for making this excellent point, and we agree that our “gamma” terminology could be more clear. In general, our manuscript concerns high gamma, though in response to reviews we have extended our analyses to additional frequencies, now covering 30-120 Hz. We show that our general finding of high-frequency asynchrony is present even in the typical gamma range of 30-60 Hz (Fig. 3). Furthermore, in this range, we identify electrodes with oscillatory signal and construct networks specific to those oscillations, helping to distinguish effects arising from broad vs. narrow-band high-frequency activity (Fig. 2). In the revised manuscript, we have also adopted use of “high gamma” to refer to spectral power assessed in the 45+ Hz range, and when we did not explicitly test for oscillations. We hope this change helps clarify the distinction between activity arising from different underlying processes.

2. *I had difficulty understanding some aspects of the analysis. My understanding is as follows: each trial has a sequence of 12 words, followed by delay and retrieval. During retrieval, some of those 12 words are remembered and others not. Each experiment has up to 25 trials (up to 300 words). Out of this full set of ~300 words, the phase difference distribution is computed for remembered and non-remembered words separately to get vectors R1 and R2, from which the F-statistic is computed. Is this*

correct? If so, then when the author's say "remembered trials" (say at line 613), they mean "remembered words", right? Or am I missing something?

We apologize for the confusing terminology -- here, "trial" was taken to mean an individual word. However, we recognize that some may read "trial" to typically mean a 12-item list of words. We have made minor modifications in the methods section to make this clear.

3. *Related to the previous point, the number of vectors used for obtaining R1 and R2 is critical, but it is not described very clearly. It would be nice if some statistics are provided regarding the percentage of correct recalls in the population, as well as the number of trials performed (mean +/- SEM), so that we know the number of trials of each condition (n1 and n2 in the F statistic formula).*

We agree that general reports of behavioral data would be valuable here. We have included a supplemental figure, shown below, which reports the total number of trials in the dataset and a breakdown by the number remembered vs. not-remembered.

Fig. 5. Behavioral data in the free-recall task. A: Accuracy distribution (n=294), mean indicated by dashed line. B: Distribution of total trials (individual word presentations) completed by each subject, mean indicated by dashed line.

However, we also note (in agreement with this reviewer's next comment), that the nonparametric permutation procedure accounts for unequal vectors; null distributions are constructed according to the number of remembered and not-remembered words in a given subject, capturing the bias present when the data in each condition is lopsided. True statistics are compared to this null distribution to generate a z-score or p-value.

4. *The resultant vector length (R) critically depends on the number of trials. It is unclear how the multipliers used in the F-statistic equation resolve this issue. Some pointers should be provided why the F-statistic was chosen the way it was. Because R1 and R2 are already normalized between 0 and 1, I do not understand why the vector lengths are divided, because large vector lengths in R1 would lead to potentially very large values of the F-statistic. For example, why are the vector lengths not subtracted instead (after correcting for the bias due to unequal number of trials)? I do believe that the randomization test takes care of most of these issues (including potential biases due to unequal number of trials), but some pointers as to why the statistic was chosen in this particular way would be very useful.*

The reviewer makes several excellent points here. We have taken these suggestions and implemented them throughout the revision. Now, we simply subtract the resultant vector lengths, as suggested (see Methods for details). The reviewer is also correct that the shuffling procedure accounts for the sample size bias in vector lengths, since the sample size for each surrogate group in the null distribution is the same as remembered vs.

not-remembered groups in the unshuffled data. Therefore, the null distribution of vector length differences may not be centered at zero, if the sample sizes are sufficiently different. The methods section has been modified for clarity.

5. *Line 692: “we sum the number of significant time-frequency points”. What is being summed? The total number of significant points, or the unthresholded z-scores as mentioned on Line 115?*

In this case, the number of significant points were being summed, and compared to the number expected by chance. However, to simplify our methods, we now simply average the memory-related phase synchrony effect across the encoding interval for each frequency band, and compare the averaged z-score to the distribution expected by chance to generate a p-value.

6. *How important is bipolar referencing for these results? It could be the case that gamma synchronizes over smaller regions, but bipolar subtraction takes out the common component. The differences shown here could be just due to different space constants associated with gamma and theta networks. I understand that re-doing the analysis for the entire dataset could be difficult, but can the authors compare the results for unipolar versus bipolar referencing for at least a partial dataset to test whether the results hold across referencing techniques?*

The reviewer raises a good point. We chose to use a bipolar montage for several reasons, outlined in response to question 1 from reviewer 1 (see above, pg. 1). Owing to the relatively sparse sampling of the brain in individual subjects, it is difficult to draw strong conclusions from a small subset of the data, and we prefer the bipolar montage for accurate analysis of local spectral power. We strongly agree that future work should seek to resolve how montage choice affects these results, noting that common average and bipolar references both have inadequacies (see Nunez et al., 1997; Schiff, 2005; Guevara, et al. 2005). Additionally, our new analysis of oscillatory synchronization (Fig. 2) demonstrates it is possible to detect high-frequency synchronization in our bipolar paradigm.

7. *The evoked response (average of the ECoG time series traces) often has power at low frequencies, especially in the theta range. Therefore, regions that produce an evoked response might show a corresponding theta-band power/synchrony. It would be interesting to see whether the results hold for the induced theta and gamma as well (obtained by first subtracting the evoked response from each trace). This could also be done on a subset of data if re-doing the analysis on 300 subjects is difficult.*

We reinterpret the reviewer’s question as follows: Is it possible that there is an evoked response occurs preferentially in remembered as compared to not-remembered trials, and if so, could the power in that low-frequency response drive the findings of enhanced theta synchrony? In response to reviewer 2, question 8 (pg. 8), we note several findings that are inconsistent with this hypothesis: (1) There is a significant ($P < 0.01$) level of memory-related theta synchrony even between 1200-1600ms after onset of the word – well beyond the timeframe for typical ERPs. (2) Our results generalize to the retrieval period, in which there is no stimulus at all. (3) A strong low-frequency stimulus locking effect, preferentially present in the remembered vs. not-remembered condition, would manifest as increased low-frequency spectral power at those frequencies. In fact, consistent with prior literature, we find a widespread decrease in low-frequency power shortly after word onset (see Fig. 3A).

Not shown here, we performed an additional analysis along the lines of what was suggested by this reviewer. Specifically, if we are interested in demonstrating that enhanced theta synchrony in a subsequent memory contrast is due to evoked potentials, we should subtract the average “remembered” trace from all remembered trials, and subtract the average “not-remembered” trace from all not-remembered trials. However, we noted that this intervention has the effect of enhancing the observed theta synchrony effect, likely because correlated

structure is being introduced to all remembered-trials and not not-remembered trials. We agree that future studies could consider the question of stimulus-locking with more sophisticated methods.

References

- Axmacher, N. et al., 2006. Memory formation by neuronal synchronization. *Brain Research Reviews*, 52(1), pp.170–182.
- Burke, J.F. et al., 2014. Human intracranial high-frequency activity maps episodic memory formation in space and time. *NeuroImage*, 85, pp.834–843.
- Burke, J.F. et al., 2013. Synchronous and asynchronous theta and gamma activity during episodic memory formation. *The Journal of neuroscience : the official journal of the Society for Neuroscience*, 33(1), pp.292–304. Available at: <http://www.ncbi.nlm.nih.gov/pubmed/23283342> [Accessed September 11, 2016].
- Ezzyat, Y. et al., 2017. Direct Brain Stimulation Modulates Encoding States and Memory Performance in Humans. *Current Biology*, 27(9), pp.1251–1258. Available at: <http://www.ncbi.nlm.nih.gov/pubmed/28434860> [Accessed July 8, 2017].
- Guevara, R. et al., 2005. Phase Synchronization Measurements Using Electroencephalographic Recordings: What Can We Really Say About Neuronal Synchrony? *Neuroinformatics*, 3(4), pp.301–314. Available at: <http://link.springer.com/10.1385/Nl:3:4:301> [Accessed June 29, 2017].
- Hughes, A.M. et al., 2012. BOSC: A better oscillation detection method, extracts both sustained and transient rhythms from rat hippocampal recordings. *Hippocampus*, 22(6), pp.1417–1428. Available at: <http://doi.wiley.com/10.1002/hipo.20979> [Accessed June 29, 2017].
- Kragel, J.E. et al., 2017. Similar patterns of neural activity predict memory function during encoding and retrieval. *NeuroImage*, 155, pp.60–71. Available at: www.elsevier.com/locate/neuroimage [Accessed July 8, 2017].
- Merkow, M.B. et al., Prestimulus Theta in the Human Hippocampus Predicts Subsequent Recognition But Not Recall. Available at: <http://memory.psych.upenn.edu/files/pubs/MerkEtal14.pdf> [Accessed July 8, 2017].
- Nunez, P.L. et al., 1997. EEG coherency. I: Statistics, reference electrode, volume conduction, Laplacians, cortical imaging, and interpretation at multiple scales. *Electroencephalography and clinical neurophysiology*, 103(5), pp.499–515. Available at: <http://www.ncbi.nlm.nih.gov/pubmed/9402881> [Accessed June 29, 2017].
- Raghavachari, S. et al., 2001. Gating of human theta oscillations by a working memory task. *The Journal of neuroscience : the official journal of the Society for Neuroscience*, 21(9), pp.3175–83. Available at: <http://www.ncbi.nlm.nih.gov/pubmed/11312302> [Accessed July 8, 2017].
- Schiff, S.J., 2005. Dangerous phase. *Neuroinformatics*, 3(4), pp.315–8. Available at: <http://www.ncbi.nlm.nih.gov/pubmed/16284414> [Accessed June 29, 2017].
- van Vugt, M.K., Sederberg, P.B. & Kahana, M.J., 2007. Comparison of spectral analysis methods for characterizing brain oscillations. *Journal of Neuroscience Methods*, 162(1–2), pp.49–63. Available at: <http://linkinghub.elsevier.com/retrieve/pii/S0165027006006042> [Accessed June 29, 2017].
- Whitten, T.A. et al., 2011. A better oscillation detection method robustly extracts EEG rhythms across brain state changes: The human alpha rhythm as a test case. *NeuroImage*, 54(2), pp.860–874. Available at: <http://www.sciencedirect.com/science/article/pii/S1053811910011614> [Accessed May 29, 2017].

Reviewers' comments:

Reviewer #1 (Remarks to the Author):

The authors have been very responsive to reviewer concerns and the manuscript is significantly improved. In fact, I worry that the authors have been over-responsive to reviewer 2, who, in my view, goes to great lengths to rescue the "gamma synchronization" hypothesis, which has largely been abandoned in the literature anyway, particularly in the context of memory. I think the authors should retain greater focus on the mechanistic role of low-frequency oscillations in coordinating memory encoding and retrieval. While refuting the gamma binding hypothesis is important, I worry that the paper is now overfocused on this issue, at the expense of the more interesting story of the role of low-frequency oscillations in synchronizing disparate hubs. I strongly disagree with the reviewer's assertion that the work is in anyway incremental. Just the opposite, while there are hints of theta synchrony-local gamma desynchrony in past work, this paper is a tour de force, bringing together multiple threads (encoding, retrieval, multiple frequency bands, analysis of both oscillations and broad band activity, all in humans) in a way that no other paper previously has even come close to. Finally, I disagree that the work does not detect oscillatory activity. While the authors have now included a method that specifically detects this, in general, wavelets will do a good job at catching mostly oscillatory activity anyway (see a recent book by Michael X Cohen that touches on this topic.)

To summarize, the paper is in excellent shape overall, although I worry reviewer 2 has pushed them too far off their original focus on low-frequency oscillations. In my view, this focus should be retained, at least to some extent, without the need to overfocus on gamma. On another note, I do agree that testing all frequencies is important, but not at the expense of a focus in particular on theta and gamma, long held as important oscillations in cognition.

I have some minor suggestions for revision:

1) "We found that the same network-level patterns of connectivity hold true in the retrieval contrast compared to the encoding contrast (Fig. 6B)."

Should be "held true"

2) "reflects the aggregation of fast, stochastic spiking activity of a population of neurons"

avoid overstatement, "reflects, in part..."

(as clearly shown in almost all work on the LFP, it is mixture of synaptic activity and action potentials, and even high gamma is not purely action potentials)

Reviewer #2 (Remarks to the Author):

The revised manuscript and the point-by-point response do not address several major concerns. Therefore, the manuscript is not suitable for publication.

The authors argue that Burke et al. 2013 aggregated over lobes, leaving open the possibility that HG synchronization takes place on finer anatomical scales. In their present manuscript, the authors move from lobe-wise interactions to ROI-wise interactions. However, those ROIs are still far too large to address this point. The relevant anatomical scales are probably columns, e.g. columns activated during task performance versus neighboring columns that are not activated.

The authors now present an analysis, in which they first search for electrodes with oscillations in the gamma band and then find enhanced synchronization during memory. They argue that these oscillations are extremely rare. However, the more likely explanation is that these oscillations are extremely hard to find with the extremely coarse iEEG recordings used.

My first review criticized that the authors do not demonstrate the presence of the rhythms, which they claim to investigate, and I made suggestions about how to demonstrate them. This explicitly pertained to both theta and gamma. The authors respond by pointing to the gamma analysis. However, there seems to be still no demonstration of the theta rhythm as a clear peak in a spectrum. Given the central role of theta in this manuscript, it is indispensable to demonstrate the presence of a clear theta rhythm.

My first review requested the authors to investigate spectra for the presence of alpha and/or beta, and to investigate memory effects if those rhythms are found. Rather than following this request, the authors analyze two more frequency ranges, without demonstrating rhythmicity through spectral peaks, and without making meaningful selections of electrodes for the presence of rhythmic activity.

My first review explained that the power in the (high) gamma range in iEEG electrodes is strongly contaminated by spikes and/or postsynaptic potentials. This precludes the interpretations that the authors try to derive. Rather, increases in this „high-gamma“ power reflect local neuronal activation and lead to decreases in long-range „high-gamma“ synchronization through a trivial mechanism. Rather than acknowledging this, the authors reiterate that their iEEG recordings obtain truly rhythmic gamma oscillations in a minority of sites. This response completely fails to address my point. This point is the most central problem of this manuscript and it remains fully un-addressed.

My first review also explained that the broadband high-frequency power changes are not gamma, because they are not a rhythm. The authors seem to agree to that. However, they still address their power changes as gamma, rather than broadband high-frequency power (or similar), throughout the paper.

The authors now state that eye movements were not controlled. That is a major problem. Contrary to the statement of the reviewers, it is well possible that differences in eye movements are the true cause of differences in neuronal activity and synchronization. The

authors argue that they find differences during retrieval. However, eye movements might differ between successful and unsuccessful retrieval, and as the patients were not in complete darkness, these eye movement differences lead to different visual input and to differences in brain-wide neuronal activity and synchronization.

Reviewer #3 (Remarks to the Author):

Review of Solomon et al.

The authors have made significant changes to address the concerns raised by all the reviewers. I think the manuscript is in much better shape, which I recommend for publication.

I have only one comment regarding the tone of the paper. Presently it is being pitched as a "theta-gamma" story, to align with a large body of work that has looked into theta and gamma oscillations in the context of learning and memory, especially in the hippocampus. However, it is quite clear that what is being addressed here are broadband responses, one in the low frequency from 0-30 Hz that includes alpha and beta as well, and another from 30 to 120 and above, including gamma and high-gamma. There is no demonstration of any theta peak, and the gamma peak shows up in only ~1% of cases. I think the work is very impressive and can easily hold its own without being pitched as a theta-gamma paper. If anything, the authors should take this opportunity to highlight how important broadband responses really are. The authors have made these points somewhat in the introduction and discussion, but they are in a position to make a more direct and much deeper impression. Having said that, I understand that the way results are presented are to some extent dependent on personal choice and beliefs, and the authors are certainly entitled to keep the present tone if they feel like that.

Reviewer #4 (Remarks to the Author):

Opinion on reviews of NCOMMs-17-04695A, Solomon et al.

I was asked to provide my opinions on the previous reviews and the responses provided by the authors in their revision to help arbitrate a disagreement between reviewers. As such, I don't provide an additional review myself but instead offer my take on the previous reviewer comments.

I carefully read the revised manuscript, the original and revision reviewer comments and authors replies. In my opinion, this is an excellent manuscript and the authors have been very responsive to reviewer comments. The dataset that the analysis is based on is of unprecedented size, allowing an unprecedented detail of anatomical precession. The results are rigorous, novel and of wide interest. Methods and statistics are well done, and the bipolar referencing used is appropriate for the purpose it is used for. The task and strategy

(subsequent memory effects) used is well suited to address brain-wide phenomena. The overall finding that wide-spread theta-band synchronization and gamma-band desynchronization predicts success of memory formation/recall and that levels of synchrony are indicative of power changes will be of wide interest to many in the field. In conclusion, I recommend this manuscript be published.

Major issues:

1. The principal disagreement concerns whether gamma-band activity is an oscillation vs. a broad-band increase due to asynchronous underlying synaptic activity increases. This is clearly an important question, and the authors have added new analysis in revision to address this for gamma oscillations. However, this issue is an entirely separate question from the one the authors address in this paper. Even if activity is broadband (and not an oscillation), this does not mean that it cannot synchronize/desynchronize. This is the question the authors address here. See, for example, the work of Burns et al 2011 J Neurosci and Burns et al 2010 J Neurosci, which showed gamma-band activity in V1 recorded with microelectrodes is "filtered noise" (=broadband activity) rather than a pure oscillation. However, for two areas to communicate, it is not required that the synchronous activity be an oscillation. It is also not required that they are visible as a clear "peak" in the power spectrum. Even if broadband and not autocorrelated, it can still be synchronous (i.e. Roberts et al Neuron 2013, Nikolic et al 2013 TICS). Thus, while both questions are important, they are separate. Properly clarifying this point in discussion is sufficient for purposes of this paper.

2. A second issue raised is that of ECOG vs. LFP. What is recorded here is ECOG, which are relatively large low-impedance contacts that integrate over a large area of the brain. In contrast, most theories of the role of gamma synchrony rely on micro-electrode recordings performed with high-impedance electrodes used for single-neuron recordings. These electrodes yield a proper LFP, whereas here the signal that is analyzed is an ECOG/iEEG signal. So these results do not speak to the possibility of the existence and synchronization of gamma-band oscillations when measured at the level of LFPs, but they do argue against this at the level of ECOG. This is entirely valid and it is sufficient to properly lay out this distinction more explicitly in the discussion.

Minor issues:

1. In several places, methods are justified because they are "popular". This is an odd word choice.

2. Throughout the manuscript, when the authors talk about increased or decreased synchronization what is meant is a relative increase, i.e. later recalled vs. not recalled. This is clear to me, but in several instances the wording could have been more clear to point out that this is the case. For example, in discussion, it is reported that "...human cognition is overwhelmingly associated with asynchronous ...". But what is overwhelming associated with less synchrony is successful memory formation relative to unsuccessful memory formation, not "human cognition". Since the analysis is a comparison between conditions, it is possible that two areas are highly synchronous in gamma but that this level of synchrony did not differ as a function of the success of encoding. This would be reported as "no synchrony". This strategy is well justified, but in some instances I recommend to clarify wording to make

this clear.

3. Reviewer 2 raises issue of not controlling for differences in eye movements. It is in my opinion not possible that eye movement differences create the widespread synchronization differences observed, particularly given the bipolar montage used (i.e. Kovach et al 2011, Neuroimage) and the small stimuli (words).

Responses to Reviewer Comments

We address each reviewer's comments below. All references to figures refer to those included in the article main text or supplementary materials.

Reviewer #1 (Remarks to the Author):

The authors have been very responsive to reviewer concerns and the manuscript is significantly improved. In fact, I worry that the authors have been over-responsive to reviewer 2, who, in my view, goes to great lengths to rescue the "gamma synchronization" hypothesis, which has largely been abandoned in the literature anyway, particularly in the context of memory. I think the authors should retain greater focus on the mechanistic role of low-frequency oscillations in coordinating memory encoding and retrieval. While refuting the gamma binding hypothesis is important, I worry that the paper is now overfocused on this issue, at the expense of the more interesting story of the role of low-frequency oscillations in synchronizing disparate hubs. I strongly disagree with the reviewer's assertion that the work is in anyway incremental. Just the opposite, while there are hints of theta synchrony-local gamma desynchrony in past work, this paper is a tour de force, bringing together multiple threads (encoding, retrieval, multiple frequency bands, analysis of both oscillations and broad band activity, all in humans) in a way that no other paper previously has even come close to. Finally, I disagree that the work does not detect oscillatory activity. While the authors have now included a method that specifically detects this, in general, wavelets will do a good job at catching mostly oscillatory activity anyway (see a recent book by Michael X Cohen that touches on this topic.)

To summarize, the paper is in excellent shape overall, although I worry reviewer 2 has pushed them too far off their original focus on low-frequency oscillations. In my view, this focus should be retained, at least to some extent, without the need to overfocus on gamma. On another note, I do agree that testing all frequencies is important, but not at the expense of a focus in particular on theta and gamma, long held as important oscillations in cognition.

I have some minor suggestions for revision:

1) *"We found that the same network-level patterns of connectivity hold true in the retrieval contrast compared to the encoding contrast (Fig. 6B)."*

Should be "held true"

2) *"reflects the aggregation of fast, stochastic spiking activity of a population of neurons"*

avoid overstatement, "reflects, in part..."

(as clearly shown in almost all work on the LFP, it is mixture of synaptic activity and action potentials, and even high gamma is not purely action potentials)

We thank the reviewer for these helpful comments. We have made several modifications to the Introduction and Discussion section to refocus the narrative on the mechanistic role of low-frequency synchronization in memory processes. Additionally, both minor points have been addressed in the revised manuscript; the noted line in the Discussion now reads "...largely reflects the aggregation of..."

Reviewer #2 (Remarks to the Author):

The revised manuscript and the point-by-point response do not address several major concerns. Therefore, the manuscript is not suitable for publication.

The authors argue that Burke et al. 2013 aggregated over lobes, leaving open the possibility that HG synchronization takes place on finer anatomical scales. In their present manuscript, the authors move from lobe-wise interactions to ROI-wise interactions. However, those ROIs are still far too large to address this point. The relevant anatomical scales are probably columns, e.g. columns activated during task performance versus neighboring columns that are not activated.

The authors now present an analysis, in which they first search for electrodes with oscillations in the

gamma band and then find enhanced synchronization during memory. They argue that these oscillations are extremely rare. However, the more likely explanation is that these oscillations are extremely hard to find with the extremely coarse iEEG recordings used.

My first review criticized that the authors do not demonstrate the presence of the rhythms, which they claim to investigate, and I made suggestions about how to demonstrate them. This explicitly pertained to both theta and gamma. The authors respond by pointing to the gamma analysis. However, there seems to be still no demonstration of the theta rhythm as a clear peak in a spectrum. Given the central role of theta in this manuscript, it is indispensable to demonstrate the presence of a clear theta rhythm.

My first review requested the authors to investigate spectra for the presence of alpha and/or beta, and to investigate memory effects if those rhythms are found. Rather than following this request, the authors analyze two more frequency ranges, without demonstrating rhythmicity through spectral peaks, and without making meaningful selections of electrodes for the presence of rhythmic activity.

My first review explained that the power in the (high) gamma range in iEEG electrodes is strongly contaminated by spikes and/or postsynaptic potentials. This precludes the interpretations that the authors try to derive. Rather, increases in this „high-gamma“ power reflect local neuronal activation and lead to decreases in long-range „high-gamma“ synchronization through a trivial mechanism. Rather than acknowledging this, the authors reiterate that their iEEG recordings obtain truly rhythmic gamma oscillations in a minority of sites. This response completely fails to address my point. This point is the most central problem of this manuscript and it remains fully un-addressed.

My first review also explained that the broadband high-frequency power changes are not gamma, because they are not a rhythm. The authors seem to agree to that. However, they still address their power changes as gamma, rather than broadband high-frequency power (or similar), throughout the paper.

The authors now state that eye movements were not controlled. That is a major problem. Contrary to the statement of the reviewers, it is well possible that differences in eye movements are the true cause of differences in neuronal activity and synchronization. The authors argue that they find differences during retrieval. However, eye movements might differ between successful and unsuccessful retrieval, and as the patients were not in complete darkness, these eye movement differences lead to different visual input and to differences in brain-wide neuronal activity and synchronization.

Reviewer #3 (Remarks to the Author):

The authors have made significant changes to address the concerns raised by all the reviewers. I think the manuscript is in much better shape, which I recommend for publication.

I have only one comment regarding the tone of the paper. Presently it is being pitched as a “theta-gamma” story, to align with a large body of work that has looked into theta and gamma oscillations in the context of learning and memory, especially in the hippocampus. However, it is quite clear that what is being addressed here are broadband responses, one in the low frequency from 0-30 Hz that includes alpha and beta as well, and another from 30 to 120 and above, including gamma and high-gamma. There is no demonstration of any theta peak, and the gamma peak shows up in only ~1% of cases. I think the work is very impressive and can easily hold its own without being pitched as a theta-gamma paper. If anything, the authors should take this opportunity to highlight how important broadband responses really are. The authors have made these points somewhat in the introduction and discussion, but they are in a position to make a more direct and much deeper impression. Having said that, I understand that the way results are presented are to some extent dependent on personal choice and beliefs, and the authors are certainly entitled to keep the present tone if they feel like that.

We have carefully considered the “broadband” angle of this manuscript, particularly in light of additional analyses we undertook in the first round of revisions. And we wholeheartedly agree that these results speak to broadband low- and high-frequency dynamics beyond the typical theta/gamma boundaries. At the same time, as Reviewer 1 noted, there is value in retaining a focus on oscillations that have been classically implicated in cognition. We have therefore sought to strike a balance between these two ideas while not diluting the paper’s

story. Admittedly, the original focus on the paper as a “theta/gamma” story still comes through, though we have made several revisions to the Introduction and Discussion to highlight the broadband nature of our observed synchrony dynamics.

Reviewer #4 (Remarks to the Author):

Opinion on reviews of NCOMMs-17-04695A, Solomon et al.

I was asked to provide my opinions on the previews reviews and the responses provided by the authors in their revision to help arbitrate a disagreement between reviewers. As such, I don't provide an additional review myself but instead offer my take on the previews reviewer comments.

I carefully read the revised manuscript, the original and revision reviewer comments and authors replies. In my opinion, this is an excellent manuscript and the authors have been very responsive to reviewer comments. The dataset that the analysis is based on is of unprecedented size, allowing an unprecedented detail of anatomical precession. The results are rigorous, novel and of wide interest. Methods and statistics are well done, and the bipolar referencing used is appropriate for the purpose it is used for. The task and strategy (subsequent memory effects) used is well suited to address brain-wide phenomena. The overall finding that wide-spread theta-band synchronization and gamma-band desynchronization predicts success of memory formation/recall and that levels of synchrony are indicative of power changes will be of wide interest to many in the field. In conclusion, I recommend this manuscript be published.

Major issues:

1. The principal disagreement concerns whether gamma-band activity is an oscillation vs. a broadband increase due to asynchronous underlying synaptic activity increases. This is clearly an important question, and the authors have added new analysis in revision to address this for gamma oscillations. However, this issue is an entirely separate question from the one the authors address in this paper. Even if activity is broadband (and not an oscillation), this does not mean that it cannot synchronize/desynchronize. This is the question the authors address here. See, for example, the work of Burns et al 2011 J Neurosci and Burns et al 2010 J Neurosci, which showed gamma-band activity in V1 recorded with microelectrodes is “filtered noise” (=broadband activity) rather than a pure oscillation. However, for two areas to communicate, it is not required that the synchronous activity be an oscillation. It is also not required that they are visible as a clear “peak” in the power spectrum. Even if broadband and not autocorrelated, it can still be synchronous (i.e. Roberts et al Neuron 2013, Nikolic et al 2013 TICS). Thus, while both questions are important, they are separate. Properly clarifying this point in discussion is sufficient for purposes of this paper.

We agree that this is an important point that was not explicitly addressed in the original manuscript. We have modified the Discussion section to make this point, in particular (pg. 13):

“[Our finding] refutes the notion that this kind of broadband signal synchronized across long distances during cognitive operations, though such interactions may still be at play in visual areas (ref. 36) and at smaller spatial scales.”

2. A second issue raised is that of ECOG vs. LFP. What is recorded here is ECOG, which are relatively large low-impedance contacts that integrate over a large area of the brain. In contrast, most theories of the role of gamma synchrony rely on micro-electrode recordings performed with high-impedance electrodes used for single-neuron recordings. These electrodes yield a proper LFP, whereas here the signal that is analyzed is an ECOG/iEEG signal. So these results do not speak to the possibility of the existence and synchronization of gamma-band oscillations when measured at the level of LFPs, but they do argue against this at the level of ECOG. This is entirely valid and it is sufficient to properly lay out this distinction more explicitly in the discussion.

This issue is also critical to the paper's main findings, and we appreciate the reviewer raising it. We have expounded on our original discussion of this point on pages 14 and 15, in which we note a discrepancy between reports that rely on microelectrode versus macroelectrode recordings. We hope our modifications in

this section make clear the distinction between activity detected at different anatomic scales, and how our results speak specifically to ECoG-observable dynamics.

Minor issues:

1. In several places, methods are justified because they are “popular”. This is an odd word choice.

These instances have been corrected in the revised manuscript.

2. Throughout the manuscript, when the authors talk about increased or decreased synchronization what is meant is a relative increase, i.e. later recalled vs. not recalled. This is clear to me, but in several instances the wording could have been more clear to point out that this is the case. For example, in discussion, it is reported that “...human cognition is overwhelmingly associated with asynchronous ...”. But what is overwhelming associated with less synchrony is successful memory formation relative to unsuccessful memory formation, not “human cognition”. Since the analysis is a comparison between conditions, it is possible that two areas are highly synchronous in gamma but that this level of synchrony did not differ as a function of the success of encoding. This would be reported as “no synchrony”. This strategy is well justified, but in some instances I recommend to clarify wording to make this clear.

We agree that readers should be reminded of the fact that our results reflect relative synchronization between two conditions. The reviewer is correct that our results do not speak to the “baseline” level of synchronization, only how synchronization changes between memory conditions. We have made several modifications throughout the text to highlight this distinction.

3. Reviewer 2 raises issue of not controlling for differences in eye movements. It is in my opinion not possible that eye movement differences create the widespread synchronization differences observed, particularly given the bipolar montage used (i.e. Kovach et al 2011, Neuroimage) and the small stimuli (words).